# Knowledge Starts with Practice: Knowledge-Aware Exercise Generative Recommendation with Adaptive Multi-Agent Cooperation

**Yangtao Zhou[1], Hua Chu[1], Yongxiang Chen[1], Ziwen Wang[1], Jiacheng Liu[1], Jianan Li[1]***
**Yueying Feng[2], Xiangming Li[1], Zihan Han[1], Qingshan Li[1]**
[1]Xidian University, [2]Zhejiang University
{zhou_yt, wawa, dsz, liu_jc, xiangmli, hanzihan}@stu.xidian.edu.cn
{lijianan}@xidian.edu.cn
{hchu, qshli}@mail.xidian.edu.cn
{yueyingf}@zju.edu.cn

## Abstract

Adaptive learning, which requires the in-depth understanding of students' learning processes and rational planning of learning resources, plays a crucial role in intelligent education. However, how to effectively model these two processes and seamlessly integrate them poses significant implementation challenges for adaptive learning. As core learning resources, exercises have the potential to diagnose students' knowledge states during the learning processes and provide personalized learning recommendations to strengthen students' knowledge, thereby serving as a bridge to boost student-oriented adaptive learning. Therefore, we introduce a novel task called Knowledge-aware Exercise Generative Recommendation (KEGR). It aims to dynamically infer students' knowledge states from their past exercise responses and customizably generate new exercises. To achieve KEGR, we propose an adaptive multi-agent cooperation framework, called ExeGen, inspired by the excellent reasoning and generative capabilities of LLM-based AI agents. Specifically, ExeGen coordinates four specialized agents for supervision, knowledge state perception, exercise generation, and quality refinement through an adaptive loop workflow pipeline. More importantly, we devise two enhancement mechanisms in ExeGen: 1) A human-simulated knowledge perception mechanism mimics students' cognitive processes and generates interpretable knowledge state descriptions via demonstration-based In-Context Learning (ICL). In this mechanism, a dual-matching strategy is further designed to retrieve highly relevant demonstrations for reliable ICL reasoning. 2) An exercise generation-adversarial mechanism collaboratively refines exercise generation leveraging a group of quality evaluation expert agents via iterative adversarial feedback. Finally, a comprehensive evaluation protocol is carefully designed to assess ExeGen. Extensive experiments on real-world educational datasets and a practical deployment in college education demonstrate the effectiveness and superiority of ExeGen. The code is available at `https://github.com/dsz532/exeGen`.

## 1 Introduction

Adaptive learning, which involves the in-depth understanding of students' learning processes and rational planning of learning resources, has been shown to improve learning outcomes, reduce dropout

---

*Corresponding Author

39th Conference on Neural Information Processing Systems (NeurIPS 2025).

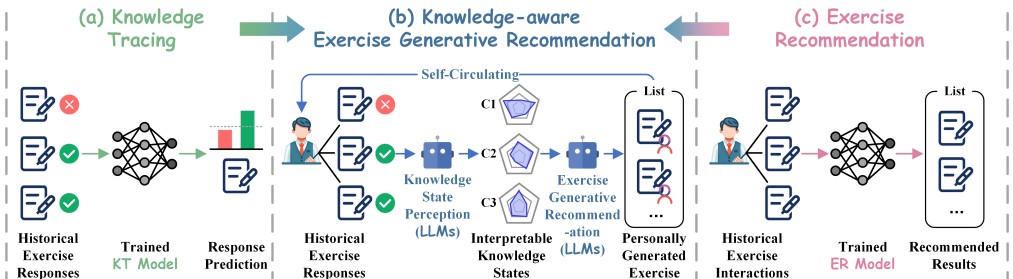

Figure 1: Comparison of three intelligent education tasks.

rates, and enhance instructor satisfaction in recent empirical studies [3, 23, 9]. Consequently, it has gained significant attention in the field of intelligent education to support personalized learning. However, how to effectively model and seamlessly integrate these two stages poses significant challenges for the implementation of adaptive learning systems. As critical learning resources, exercises play a pivotal role in both knowledge diagnosis and acquisition [36], which has the potential to diagnose students' knowledge states [27, 26] during the learning processes and provide personalized learning recommendations to strengthen students' knowledge [29, 20], thereby serving as a natural bridge to boost student-oriented adaptive learning.

Therefore, we introduce a novel task termed **K**nowledge-aware **E**xercise **G**enerative **R**ecommendation (KEGR), which leverages exercises to bridge knowledge state perception and personalized learning recommendations. As illustrated in Figure 1(b), KEGR aims to dynamically ***perceive students' knowledge states*** from their historical exercise responses, and subsequently ***generate and recommend tailored exercises*** based on the knowledge states, thereby offering a personalized learning experience. The most related tasks include Knowledge Tracing (KT) and Exercise Recommendation (ER):

- KT (Figure 1(a)) typically predicts whether a student can answer exercises correctly based on the past responses [63, 1], but fails to provide interpretable knowledge state descriptions for downstream applications and often overlooks the semantics within exercises and concepts.

- ER (Figure 1(c)) usually applies general recommendation methods to match students with exercises drawn from a static, one-size-fits-all exercise pool, but ignores the impact of the knowledge states, which may lead to ineffective recommendations [53, 32].

The inherent limitations and isolation of these two tasks hinder the realization of adaptive learning systems. KEGR addresses this by continuously perceiving students' knowledge states and customizably generating new exercise recommendations to form a unified and dynamic closed-loop system, aligning with humans' long-term learning goals [11].

Recent advances in Artificial Intelligence Generated Content (AIGC) [62], particularly Large Language Models (LLMs) [39, 62], offer a promising solution to achieve KEGR. LLMs can deeply understand semantic information and generate rich textual content [31], allowing them to infer students' knowledge states from the answered exercises and generate personalized exercises through natural language prompts. However, due to the complexity of KEGR——which involves interactive reasoning over student responses, exercises, and concepts, a single LLM struggles to achieve the task's adaptive goal. To address this, we propose leveraging the AI Agent technology [59, 49], which excels in autonomous language interaction, role-playing, and decision-making for complex tasks. This technology can effectively stimulate LLM capabilities through multi-agent cooperation to generate high-quality knowledge-aware exercise recommendations.

To this end, we propose a Knowledge-aware **Exe**rcise **Gen**erative Recommendation Framework with Adaptive Multi-a**Gen**t Cooperation, abbreviated as **ExeGen**. This framework integrates four specialized agents: *Recommendation Manager*, *Knowledge Perceiver*, *Exercise Generator*, and *Quality Evaluation Expert*, responsible for global supervision, knowledge state perception, exercise generation, and quality refinement, respectively. These agents collaborate through an adaptive loop workflow pipeline to generate personalized recommendations. Built on this framework, we devise two key mechanisms to enhance the precision of knowledge state perception and the quality of generated exercises. 1) A human-simulated knowledge perception mechanism (intra-agent) for the *Knowledge Perceiver*, mimics students' cognitive processes with a domain knowledge graph to construct high-

quality demonstrations, enabling finer-grained tracking of their thought paths. In this way, the *Knowledge Perceiver* can prompt LLMs to infer the knowledge mastery of students more accurately and generate interpretable knowledge state descriptions via demonstration-based In-Context Learning (ICL). In this mechanism, a dual-matching strategy is further designed to retrieve relevant contextual demonstrations via precise and fuzzy searches, thereby boosting perception reliability. 2) An exercise generation-adversarial mechanism (inter-agent) establishes a feedback-driven adversarial interaction between the *Exercise Generator* and *Quality Evaluation Expert*. Notably, a group of evaluation experts, each focusing on different quality aspects, collaboratively guide the *Exercise Generator* to iteratively refine the generated exercises, resulting in more accurate and pedagogically personalized exercises. To evaluate ExeGen, we design a comprehensive evaluation protocol that integrates GPT-based scoring, statistical analysis, and human evaluations from both students and teachers. Extensive experiments on real-world educational datasets and a practical deployment in college education confirm the effectiveness and superiority of ExeGen, highlighting its promise for advancing adaptive learning systems. Our main contributions are summarized as follows:

• We introduce a novel task KEGR, and propose a LLM-powered framework ExeGen that achieves personalized exercise generation and recommendation through adaptive multi-agent cooperation. By coordinating four specialized agents in a closed-loop workflow pipeline, ExeGen bridges the gap between knowledge state perception and personalized learning recommendations.

• We devise a human-simulated knowledge perception mechanism in *Knowledge Perceiver*, which enables finer-grained tracking of knowledge states through human-simulated cognitive paths, with a dual-matching strategy for further reliable inference. This approach achieves superior transparency and interpretability compared to current KT methods, advancing educational diagnostic applications.

• We design an exercise generation-adversarial mechanism between the *Exercise Generator* and *Quality Evaluation Expert*, where multi-dimensional experts collaboratively refine exercise generation through iterative adversarial feedback. This mechanism ensures that the generated exercises adhere rigorous educational standards while effectively addressing individual learning needs.

• We design a comprehensive evaluation protocol to assess ExeGen. Extensive experiments on real-world educational datasets and an actual application demonstrate the effectiveness and superiority of ExeGen.

## 2 Related Work

**Exercise Recommendation**. Recent work adapts general recommendation technologies [22, 52, 64] to educational settings for personalized learning resource recommendations [58, 19, 48, 60, 29]. Effective Exercise Recommendations (ER) help guide students along appropriate learning paths and boost engagement [53]. Early efforts formulate ER as a multi-objective optimization task. For example, Huang et al. [24] and Liu et al. [32] use deep reinforcement learning to balance review & explore, difficulty progression, and engagement. To model learning dynamics, Recurrent Neural Networks (RNNs) [53] and Long Short-Term Memory (LSTM) networks [20, 21] are employed to capture students' knowledge mastery and forgetting rates. More recently, Li et al. [29] employ Graph Neural Networks (GNNs) to capture heterogeneous relations among students, exercises, and concepts, further improving the recommendation performance. Despite these advances, most existing methods overlook the nuances of individual students' knowledge states and rely only on static exercise pools, leading to one-size-fits-all recommendations that might not align with students' personalized needs.

**Knowledge Tracing**. Knowledge Tracing (KT) aims to infer students' evolving knowledge states from their past exercise responses [63, 1]. Existing methods fall into two categories: traditional probabilistic methods and deep learning methods [54]. Early probabilistic methods, such as Item Response Theory (IRT) [18], Bayesian Knowledge Tracing (BKT) [8], and Factor Analysis Models (FAM) [5, 41, 7], use handcrafted features to predict student performance [2]. In recent years, deep learning methods aim to learn latent patterns directly from large-scale data [36, 57]. For example, Piech et al. [42] first introduce RNNs to model response sequences, enabling dynamic tracking of knowledge states. Pandey et al. [40] incorporate self-attention to capture fine-grained knowledge states. Graph-based methods further extend KT by representing concept dependencies through GNNs [38, 1, 54]. Despite these advances, current methods simplify the KT task to binary correctness prediction, which limits interpretability and reduces their potential to support personalized learning.

**LLMs for Intelligent Education**. Large Language Models (LLMs), such as GPTs, have significantly accelerated progress in intelligent education [51, 25], supporting a wide range of applications in personalized instruction, assessment, and content generation [33]. For personalized instruction, LLMs are used to deliver Socratic-style tutoring [33, 47] or to simulate interactive classroom environments through AI Agents [61]. For learning outcome assessment, LLMs help evaluate student work automatically, reducing teacher workload [10, 28]. For educational content generation, LLMs are used to produce textbooks, instructional images, and coding exercises aligned with curriculum goals [4, 16]. However, most of the existing exercise generation work centers on teacher-facing tools, overlooking the potential for adaptive, student-centered exercise generation [9, 37].

## 3 Problem Formulation

Let $\mathcal{U} = \{u_1, u_2, \ldots, u_{|\mathcal{U}|}\}$, $\mathcal{E} = \{e_1, e_2, \ldots, e_{|\mathcal{E}|}\}$, and $\mathcal{C} = \{c_1, c_2, \ldots, c_{|\mathcal{C}|}\}$ denote the sets of students, exercises, and knowledge concepts, respectively. For a given student $u_i \in \mathcal{U}$, his/her learning history is denoted as $H^{u_i} = \{(e_1^{u_i}, c_1^{u_i}, y_1^{u_i}), \ldots, (e_j^{u_i}, c_j^{u_i}, y_j^{u_i}), \ldots, (e_n^{u_i}, c_n^{u_i}, y_n^{u_i})\}$, where $n$ denotes the length of the learning history. Here, each tuple in $H^{u_i}$ consists of an exercise $e_j^{u_i} \in \mathcal{E}$, its associated knowledge concept $c_j^{u_i} \in \mathcal{C}$, and a binary label $y_j^{u_i} \in \{0, 1\}$ indicating whether the student answered correctly. The goal of the KEGR task is to generate personalized and new exercises that align with the evolving knowledge state of each student. KEGR is achieved by addressing two tightly-coupled subtasks via natural language interactions:

- **Knowledge State Perception**: Given a student $u_i$'s exercise history $H^{u_i}$, this subtask aims to generate an interpretable textual description $s_{n+1}^{u_i}$ of the student's current knowledge state, enabling clear understanding and actionable insights for exercise generation.
- **Exercise Generative Recommendation**: Based on $s_{n+1}^{u_i}$, this subtask aims to generate personalized exercises $\hat{e}_{n+1}^{u_i}$, along with their corresponding concepts $c_{n+1}^{u_i}$, aligning well with the student's learning needs.

These subtasks form an iterative feedback loop that continuously monitors the learning process of students and dynamically adjusts exercise generation, promoting long-term and personalized learning gains. For simplicity, we omit student-specific superscript $u_i$ in the remainder of the paper.

## 4 Methodologies

### 4.1 Adaptive Multi-Agent Cooperation Framework

In this section, we present ExeGen, a framework that leverages LLMs to generate personalized knowledge-aware exercise recommendations. As shown in Figure 2, ExeGen integrates four specialized agents: ***Recommendation Manager***, ***Knowledge Perceiver***, ***Exercise Generator***, and ***Quality Evaluation Expert***. These agents work collaboratively in an adaptive loop workflow pipeline that follows the process of "recommendation monitoring → knowledge perception → exercise generation → quality refinement". This design enables personalized and interpretable learning support through natural language interaction. The detailed prompts for all agents are listed in Appendix B.1.

**Recommendation Manager** is the core of ExeGen, which acts as the controller and orchestrator of the entire pipeline. Inspired by recent studies [50, 35] that task decomposition strategy can significantly enhance LLMs' problem-solving capabilities, the Recommendation Manager breaks down the complex KEGR task into three sub-steps: knowledge state perception, personalized exercise generation, and exercise quality evaluation. It receives the historical exercise response data $H$ of student $u_i$, and supervises each sub-step of the pipeline, invoking the appropriate agents in sequence. After collecting well-evaluated exercises $e_{n+1}$ from the final sub-step, it determines whether the recommendation cycle is complete and delivers the newly generated exercises to the student $u_i$. This continuous monitoring enables an adaptive, feedback-driven learning loop.

**Knowledge Perceiver** focuses on prompting LLMs to analyze the historical exercise records $H$ provided by the Recommendation Manager, thereby generating accurate and interpretable knowledge state description $s_{n+1}$ of student $u_i$. However, LLMs often struggle with accurate knowledge inference due to hallucination issues and limited reasoning depth [43]. To mitigate these challenges, we devise a human-simulated knowledge perception mechanism with a domain knowledge graph for

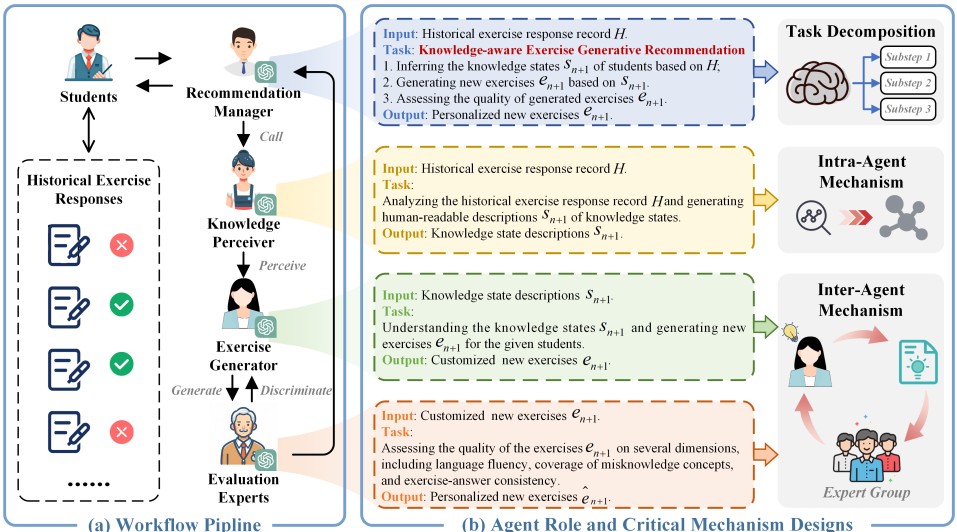

Figure 2: Overall framework of ExeGen consists of four specialized agents.

the Knowledge Perceiver. This mechanism mimics human exercise-answering behavior that involves relevant knowledge retrieval → analysis → exercise solving, enabling finer-grained tracking of the student's thought paths as well as the generation of interpretable and trustworthy descriptions of the student's knowledge states. Further details are introduced in Section 4.2.

**Exercise Generator** crafts personalized exercises tailored to the inferred knowledge states from the Knowledge Perceiver. Each exercise includes a stem, an associated concept, and a standard answer. To ensure knowledge state alignment and reasoning quality, we integrate Chain-of-Thought (CoT) [50] prompting during exercise generation. The Exercise Generator also supports prompt-level control through two configurable parameters: exercise type and quantity. Due to the subjectivity and evaluation challenges of subjective exercise [37], we only focus on three objective exercise types: single-choice, multi-choice, and judgment, leaving the subjective exercise as our future work. For exercise quantity, the default setting is 10, but can be adjusted to match the learning goals of students.

**Quality Evaluation Expert** ensures the pedagogical quality of generated exercises through a multi-dimensional, adversarial refinement process. Drawing inspiration from the generative-discriminative dynamics in Generative Adversarial Networks (GANs) [14, 6], we establish an exercise generative-adversarial mechanism between the Exercise Generator and a group of specialized Quality Evaluation Experts. Considering that exercises in educational scenarios need to satisfy multi-aspect constraints, each expert targets a distinct quality aspect, collectively forming a multi-dimensional constraint network. This network enforces the Exercise Generator to iteratively refine its outputs until they meet all quality thresholds. Section 4.3 provides further details of this mechanism.

## 4.2 Human-simulated Knowledge Perception Mechanism (Intra-Agent)

While LLMs possess extensive world knowledge and strong logical reasoning capabilities [44], they face two major challenges in knowledge state perception. First, LLMs lack domain-specific knowledge, limiting their ability to capture the hierarchical relationships among courses, concepts, and exercises, which are critical for modeling students' knowledge states [17]. Second, they are prone to hallucinations that compromise inference reliability in knowledge-intensive tasks [43]. To address these challenges, we propose a human-simulated knowledge perception mechanism that aligns LLM reasoning with the multi-step cognitive process humans follow when answering exercises. Additionally, this mechanism integrates a domain knowledge graph from a real-world dataset to enhance domain-specific knowledge retrieval and alleviate hallucinations. As shown in Figure 3, the mechanism consists of two modules: demonstration construction and knowledge state generation. The former models human cognitive steps and embeds structured domain knowledge into contextual demonstrations, enabling step-by-step thinking processes. The latter uses demonstration-based ICL

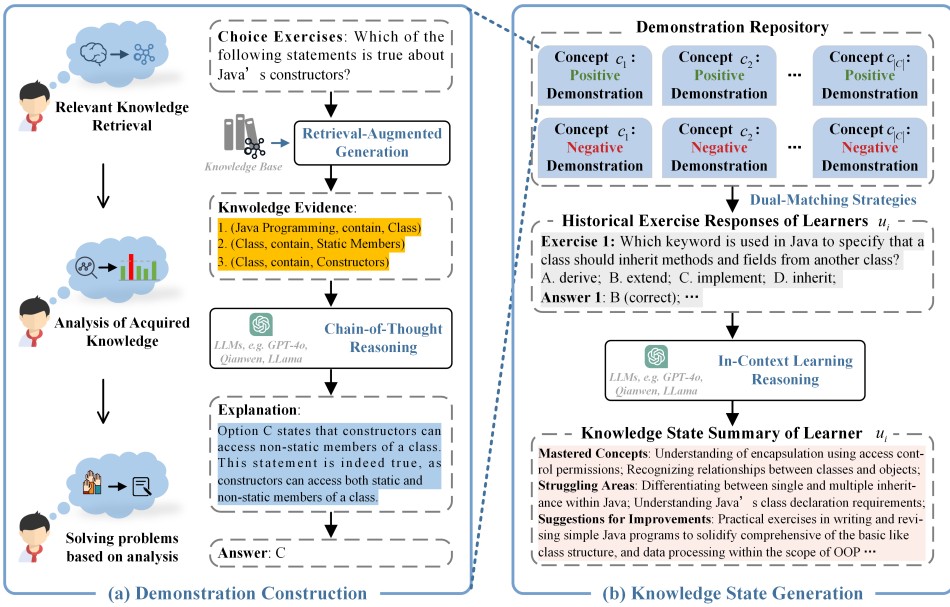

Figure 3: Human-simulated knowledge perception mechanism.

to produce intermediate reasoning steps before generating the final knowledge states. This design effectively enhances interpretability, reduces hallucinations, and improves inference accuracy.

### 4.2.1 Knowledge Retrieval-augmented Demonstration Construction

Prior work [13] shows that well-designed demonstrations can significantly enhance LLM reasoning. To this end, we propose a knowledge retrieval-augmented demonstration construction module to craft high-quality, task-specific demonstrations prior to LLMs for knowledge state generation. As shown in Figure 3(a), this module mimics the structured cognitive process humans follow when answering exercises: 1) identifying assessed concepts and retrieving relevant prior knowledge; 2) analyzing the exercises in light of the retrieved knowledge; 3) synthesizing insights to derive a solution. By simulating this process, the module generates demonstrations that comprehensively cover all the required concepts.

Firstly, the module employs Retrieval-Augmented Generation (RAG) techniques [15] to extract Knowledge Evidences (KEs) [46] from a domain-specific Knowledge Graph (KG), simulating the initial step of the human cognitive process. Each KE is a structured knowledge triple:

$$KE_i = (h_i, r_i, t_i) \,|\, i = 1, 2, \cdots, m \tag{1}$$

where $h_i$ and $t_i$ denote the head and tail entities, and $r_i$ represents their relationship in the KG. The configurable parameter $m$ controls the number of retrieved triplets, affecting the depth and breadth of relevant knowledge. These KEs encapsulate critical information, such as $(address\ mapping, is\_pre, virtual\ address)$, indicating that "$address\ mapping$" is a prerequisite concept for "$virtual\ address$".

Secondly, we apply zero-shot CoT prompting, which uses the phrase "Let's think step by step" in the prompt to guide the LLM in generating explanatory reasoning for each knowledge concept $c_i \in \mathcal{C}$ and its associated exercise. This step simulates the second step of the human cognitive process along with the retrieved KEs to reduce hallucinations, which is formally defined as:

$$\begin{cases} x_{i,pos}^{DE} = LLM\,(e_{pos}, c_i, KE_i)\,|i = 1, 2, \ldots, |\mathcal{C}| \\ x_{i,neg}^{DE} = LLM\,(e_{neg}, c_i, KE_i)\,|i = 1, 2, \ldots, |\mathcal{C}| \end{cases} \tag{2}$$

where $x_{i,pos}^{DE}$ and $x_{i,neg}^{DE}$ are the LLM-generated explanations derived from the positive and negative samples, respectively. Here, $e_{pos}$ and $e_{neg}$ represent the positive sample (exercise with correct response) and negative sample (exercise with incorrect response) for concept $c_i$, respectively. This dual-sample strategy enriches the demonstrations and avoids observational bias during reasoning.

Finally, we construct structured demonstrations by concatenating each knowledge concept $c_i$, exercise instances, student responses, KEs, and the LLM-generated explanations, which simulates the final step of the human cognitive process. These demonstrations provide contextual cues that help LLMs associate knowledge understanding with student responses. The construction process is defined as:

$$\begin{cases} DE_{i,pos} = e_{pos} \| c_i \| KE_i \| x^{DE}_{i,pos} \| y_i \\ DE_{i,neg} = e_{neg} \| c_i \| KE_i \| x^{DE}_{i,neg} \| y_i \end{cases} \tag{3}$$

where $DE_{i,pos}$ and $DE_{i,neg}$ denote the positive and negative demonstrations, with $y_i = 1$ for correct responses and $y_i = 0$ for incorrect ones. All demonstrations are stored in a demonstration repository and used for In-Context Learning (ICL) in the subsequent knowledge state generation stage.

#### 4.2.2 ICL-based Knowledge State Generation

To accurately infer the knowledge states of students, we introduce an ICL-based knowledge state generation module that leverages retrieved demonstrations to provide contextual support. As illustrated in Figure 3 (b), given a student's historical exercise responses $H$, each record $(e_j, c_j, y_j) \in H$ is paired with the top-$K$ most relevant demonstrations from the pre-built repository. Specifically, we introduce two retrieval strategies:

- **Hard matching strategy** selects demonstrations that exactly match the concept $c_j$ from the repository. If no match is found, indicating a novel or emerging concept, the module switches to the soft matching strategy.

- **Soft matching strategy** firstly encode both the input record (as a Query token) $(e_j, c_j, y_j)$ and stored demonstrations via a pre-trained language model (such as Bert [12]) to yield their respective semantic representations $v_{Query}$ and $v_{DE}$. The strategy then selects the most semantically relevant demonstrations based on their cosine similarity [56].

The retrieved demonstrations are concatenated with each exercise record, and subsequently integrated into the system prompt of the Knowledge Perceiver, thereby prompting the LLM to generate both the inferred knowledge evidence $\hat{KE}_j$ and step-by-step explanation $\hat{x}_j$ from a student perspective. Finally, the Knowledge Perceiver synthesizes the exercise-answering explanations of historical exercise records to infer the overall knowledge state of student $u_i$:

$$s_{n+1} = LLM \left( \bigcup_{j=1}^{n} \left( \hat{KE}_j, \hat{x}_j \right) \right), \tag{4}$$

where $\bigcup_{j=1}^{n} (\cdot)$ denotes successive concatenation operation. This process aligns LLM reasoning with human cognitive processes to yield a more accurate and interpretable knowledge state $s_{n+1}$.

### 4.3 Exercise Generative-Adversarial Mechanism (Inter-Agent)

The reliability of the exercises is critical for adaptive learning systems. To ensure the reliability and pedagogical quality of LLM-generated exercises, we design an exercise generative-adversarial mechanism, inspired by the adversarial idea between generators and discriminators in GANs [14, 6]. This mechanism enforces a generative-discriminative validation loop between the Exercise Generator and Quality Evaluation Experts. Moreover, considering that exercises in educational scenarios need to satisfy multi-dimensional requirements at the same time, we introduce a group of evaluation experts, where each expert focuses on a specific quality dimension: 1) linguistic fluency, 2) coverage of erroneous knowledge concepts, and 3) overall correctness and reasonableness. These dimensions are configurable. And assigning one expert per aspect improves the clarity of the assessment. Specifically, experts assess each generated exercise and provide structured feedback. If the exercise fails any standard, the feedback is incorporated into the prompt of the Exercise Generator, guiding the LLM to revise the exercise. This process continues iteratively until the exercises satisfy all quality standards or the iterative process reaches a maximum of 10 rounds. The final refined exercise $\hat{e}_{n+1}$ is then passed to the Recommendation Manager for delivery to student $u_i$.

# 5 Experiments

## 5.1 Experimental Setups

**Datasets and Baselines**. We evaluate ExeGen on the MOOCCubeX dataset [55] with a KG, and the detailed dataset statistics are provided in Appendix A.1. As KEGR is a new task, there are no existing methods for direct comparison. Therefore, we adapt four representative LLM-based approaches as baselines: Zero-shot Learning [34], In-Context Learning (ICL) [13], Chain-of-Thought (CoT) [50], and ICL+CoT. The implementations and prompts for all baselines are listed in Appendix A.2 and B.2.

**Evaluation Metrics**. To comprehensively assess ExeGen, we design a three-step evaluation protocol: 1) offline hybrid scoring, 2) online post-questionnaires from the student perspective, and 3) online expert evaluation from the teacher perspective. Motivated by prior research [30], which shows that GPT-based evaluation can align with manual evaluation on content generation tasks, we incorporate both GPT-based and statistical metrics in the offline scoring step. Specifically, we design six GPT-based metrics tailored to KEGR: Knowledge Relevance (KR), Clarity, Answer Accuracy (AA), Difficulty Appropriateness (DA), Engagement and Fun (EF), and Safety and Ethics (SE). Meanwhile, we design two statistical metrics: Error Hit Rate (HR) and Error Recall (Recall). GPT-based metrics are scored on a 0-5 scale, while statistical metrics range from 0 to 1, with higher scores indicating better performance. Detailed implementations and prompts of evaluation metrics are provided in Appendix A.3 and B.3. In addition, we develop a dual-interface system based on ExeGen for both teachers and students, which has been deployed in real-world college scenarios (see Appendix C).

## 5.2 Performance Comparison

To verify the superiority of ExeGen, we evaluate it against four baselines on three exercise types: single choice, multiple choice, and judgment. The experimental results are summarized in Table 1, with optimal results highlighted in **bold** and suboptimal results marked with an underlined. Based on the results, we find the following key conclusions: Firstly, ExeGen consistently outperforms all baselines across all exercise types, demonstrating its superior capability to capture student knowledge states and generate personalized exercise recommendations. These results further validate the effectiveness of integrating multiple specialized agents, which provide more reliable generation than direct prompt-based methods. Secondly, among the baselines, Zero-shot performs the worst, while ICL and CoT show clear improvements. This result aligns with previous findings [13, 50] that in-context demonstrations and stepwise reasoning help LLMs better understand tasks and follow instructions. Finally, ICL+CoT achieves the sub-optimal performance, surpassing both the ICL and CoT. This result highlights the complementary benefits of integrating in-context learning with stepwise reasoning for improving the logical inference and task comprehension capabilities of LLMs.

Table 1: Performance comparison of different methods.

| Types | Methods | KR | Clarity | AA | DA | EF | SE | HR | Recall |
|---|---|---|---|---|---|---|---|---|---|
| Single Choice Exercise | Zero_shot | 3.77 | 3.67 | 3.69 | 3.44 | 3.28 | 4.27 | 0.11 | 0.20 |
| | ICL | 4.01 | 4.09 | 4.13 | 4.10 | 4.01 | 4.21 | 0.29 | 0.35 |
| | COT | 4.06 | 4.38 | 3.87 | 3.91 | 3.86 | 3.82 | 0.31 | 0.40 |
| | ICL+COT | 4.07 | 4.35 | 4.51 | 4.24 | 4.09 | 4.32 | 0.19 | 0.45 |
| | ExeGen | **4.81** | **4.78** | **4.89** | **4.81** | **4.72** | **4.62** | **0.70** | **0.75** |
| Multiple Choice Exercise | zero_shot | 3.63 | 3.62 | 3.49 | 3.60 | 3.41 | 4.11 | 0.08 | 0.15 |
| | ICL | 4.07 | 3.99 | 3.99 | 4.09 | 3.82 | 4.25 | 0.19 | 0.30 |
| | COT | 4.06 | 4.25 | 4.09 | 4.05 | 3.69 | 4.02 | 0.30 | 0.35 |
| | ICL+COT | 4.25 | 4.40 | 4.39 | 4.28 | 4.11 | 4.34 | 0.21 | 0.40 |
| | ExeGen | **4.82** | **4.73** | **4.79** | **4.80** | **4.58** | **4.69** | **0.79** | **0.82** |
| Judgment Exercise | zero_shot | 3.77 | 3.63 | 3.66 | 3.29 | 3.22 | 3.83 | 0.11 | 0.18 |
| | ICL | 3.84 | 4.22 | 4.21 | 4.15 | 4.25 | 3.96 | 0.20 | 0.28 |
| | COT | 3.94 | 4.29 | 3.86 | 3.88 | 3.68 | 3.82 | 0.12 | 0.30 |
| | ICL+COT | 4.03 | 4.54 | 4.57 | 4.24 | 4.10 | 4.23 | 0.31 | 0.42 |
| | ExeGen | **4.71** | **4.61** | **4.62** | **4.80** | **4.66** | **4.63** | **0.65** | **0.72** |

## 5.3 Ablation Study

We conduct ablation studies to assess the impact of different key components by removing each component individually while keeping the others unchanged, resulting in five variants: **w/o. KP** removes the Knowledge Perceiver (KP), directly generating exercises based on students' historical responses via other agents. **w/o. EG** removes the Exercise Generator (EG) and prompts the Recommendation Manager to generate exercises directly according to the Knowledge Perceiver outputs. **w/o. QEE** removes the Quality Evaluation Experts (QEEs), presenting exercises to students without further assessment and refinement. **w/o. HKP** removes the Human-simulated Knowledge Perception mechanism (HKP), directly generating the descriptions of knowledge states in a single step based only on students' past exercise responses. **w/o. EGA** eliminates the Exercise Generative-Adversarial mechanism (EGA), only using a single evaluation expert without multi-dimensional refinement.

Table 2: The ablation performance of ExeGen on multiple choice exercises.

| Methods | KR | Clarity | AA | DA | EF | SE | HR | Recall |
|---|---|---|---|---|---|---|---|---|
| w/o. KP | 3.26 | 3.40 | 3.31 | 3.32 | 3.25 | 3.27 | 0.20 | 0.25 |
| w/o. EG | 3.70 | 3.92 | 3.73 | 3.77 | 3.98 | 3.24 | 0.29 | 0.30 |
| w/o. QEE | 4.31 | 4.21 | 4.27 | 4.14 | 4.31 | 4.13 | 0.30 | 0.35 |
| w/o. HKP | 3.53 | 3.46 | 3.50 | 3.58 | 3.50 | 3.54 | 0.20 | 0.26 |
| w/o. EGA | 4.37 | 4.42 | 4.37 | 4.38 | 4.46 | 4.37 | 0.41 | 0.40 |
| ExeGen | **4.82** | **4.73** | **4.79** | **4.80** | **4.58** | **4.69** | **0.79** | **0.82** |

Table 2 presents the performance of each variant against ExeGen in the multiple-choice exercise scenario. The results indicate that removing any component leads to a significant drop across all eight evaluation metrics, highlighting the indispensable role of each component. Among them, w/o. KP exhibits the largest performance degradation, validating the critical role of accurate knowledge state modeling in guiding effective exercise generation. w/o. QEE further emphasizes the necessity of quality control. Without rigorous assessment and refinement, the generated exercises might suffer from limited practical effectiveness. Additionally, the significant drop of w/o. HKP confirms that simulating human-like perception effectively improves the LLM's understanding for student knowledge states. Lastly, the performance drop of w/o. EGA confirms that iterative expert feedback is useful for improving exercise quality. In conclusion, these findings collectively affirm that each component is essential for generating accurate, personalized, and high-quality exercise recommendations.

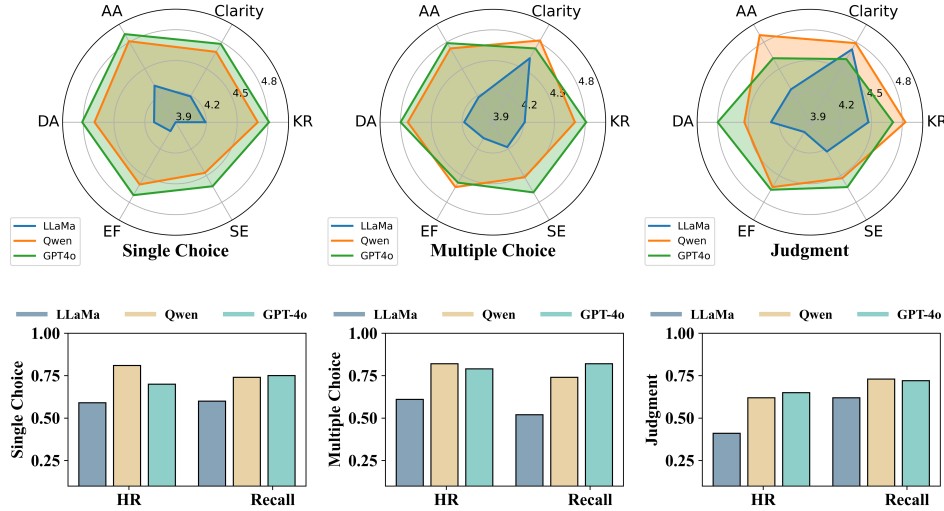

Figure 4: Influences of different LLM types.

## 5.4 Further Analysis on LLM types

We examine the impact of different base LLMs on ExeGen's performance using LLaMa-70B, Qwen-turbo-1101, and GPT-4o. As shown in Figure 4, all base models perform well across all evaluation metrics, confirming the robustness of ExeGen's knowledge state perception and iterative generation strategies. Among them, Qwen-turbo-1101 and GPT-4o achieve the strongest and comparable

performance, highlighting their advanced language understanding and reasoning capabilities. In this paper, we use GPT-4o to implement ExeGen. Besides, we further explore the impact of different LLM scales and conduct two kinds of human evaluations, which are detailed in Appendix A.5.

## 5.5 Case Study

We conduct a case study using a randomly sampled student from the MOOCCubeX dataset. As shown in Figure 5, Knowledge Perceiver produces an interpretable diagnostic report detailing mastered concepts, weak areas, and suggestions for improvement. For example, although the student answers exercises on hard and symbolic links correctly in isolation, he/she struggles with exercises combining both, indicating a gap in integrated understanding. Besides, to enhance exercise quality, we explicitly design the prompt of Exercise Generator to encourage variation in wording, difficulty, and context: "*Ensure diversity in wording, difficulty levels, and scenarios to maintain the student's engagement and provide an appropriate level of challenge*". The report of Evaluation Experts confirms that the generated exercises are well-aligned with the diagnosed weaknesses and display strong diversity. For example, exercises 3, 5, and 9 undergo iterative refinement to target different aspects of the hard links and symbolic links, including correct usage, cross-partition behavior, and conceptual limitations.

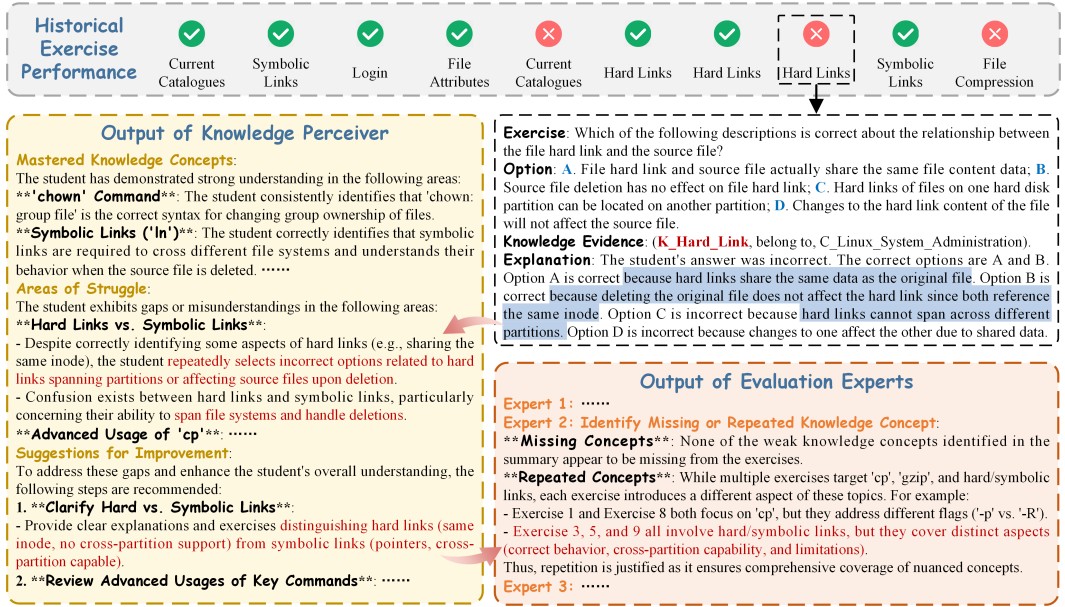

Figure 5: ExeGen generates a knowledge state report and a exercise quality evaluation report.

## 6 Conclusion

This paper introduces a novel task KEGR, which aims to perceive students' knowledge states and generate personalized exercise recommendations for supporting adaptive learning. To achieve KEGR, we propose ExeGen, a multi-agent cooperation framework built on LLMs, where four specialized agents collaborate in an adaptive loop workflow. Moreover, we devise two key mechanisms: a human-simulated knowledge perception mechanism to achieve interpretable inferences of knowledge states, and an exercise generative-adversarial mechanism to refine the generated exercise from multiple dimensions. Finally, extensive experiments on real-world datasets and an actual application confirm the effectiveness of ExeGen. The limitations and future work are detailed in Appendices D.

## Acknowledgments

This work was supported by the National Natural Science Foundation of China (U21B2015 and 62372351), Natural Science Foundation of Jiangsu Province (BK20232028), the Young Scientists Fund of the National Natural Science Foundation of China (62202356 and 62302373), Intelligent Financial Software Engineering New Technology Joint Laboratory Project (99901220858), the Fundamental Research Funds for the Central Universities (QTZX24072), and the Artificial Intelligence-Empowered Course Reform Project of Xidian University (ZNB2409).

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

# Appendices

## A    Experiment Details

### A.1    Dataset Statistics

To evaluate the effectiveness of ExeGen, we use the public MOOCCubeX dataset [55], sourced from one of China's largest MOOC platforms. Given its rich content and extensive user base, MOOCCubeX is widely used in educational data mining, particularly for online learning research. The statistical details are provided in Table 3. Following prior studies [58, 19, 48], we extract all student interaction data from the computer science and technology domain for our experiments.

Table 3: Statistics of the collected dataset.

| Entities | #Entities | Relations | #Relations |
|---|---|---|---|
| Course | 63 | Course-Concept | 52451 |
| Student | 18638 | Exercise-Concept | 4204 |
| Knowledge Concept | 27918 | Student-Course | 101007 |
| Exercise | 5015 | Student-Exercise | 6153 |

### A.2    Baseline Models

This section details the four baseline models adapted for comparison with our ExeGen framework for the Knowledge-aware Exercise Generative Recommendation (KEGR) task, all of which are implemented using the GPT-4o API interface without involving any large-model training or fine-tuning. The prompts for these baselines are provided in Appendix B.2.

- **Zero-shot Learning (Zero_shot)**: This baseline [34] uses task-specific prompts to directly generate exercise recommendations based on the LLMs' world knowledge and natural language understanding capabilities.

- **In-Context Learning (ICL)**: This baseline [13] provides LLMs with task-relevant contextual information, such as demonstrations and structured knowledge descriptions, enabling them to infer underlying patterns and generate exercises accordingly.
- **Chain-of-Thought (CoT)**: This baseline [50] decomposes the exercise generation process into intermediate inference steps, simulating the human thought process to enhance logical coherence and interpretability.
- **ICL + CoT**: This hybrid baseline combines ICL and CoT techniques. It first enriches LLMs with contextual information and then guides them through a stepwise reasoning process to generate exercise recommendations.

## A.3 Three-step Evaluation Protocol

To comprehensively assess the proposed ExeGen, we design a three-step evaluation protocol, including offline hybrid scoring, online post-questionnaires from the student perspective, and online expert evaluation from the teacher perspective.

**Step 1: Offline Hybrid Scoring**. Extensive empirical studies [30] have demonstrated a strong correlation and consistency between GPT-based scoring methods and human evaluation, making it a viable alternative for large-scale task performance assessment without requiring manual annotation. Moreover, GPT-based scoring methods exhibit adaptability to various natural language generation evaluation tasks through the design of task-specific prompts. Therefore, to systematically assess the effectiveness of our proposed ExeGen for the KEGR task, we introduce a hybrid scoring approach that combines GPT-based scoring (using advanced GPT-4o) with statistical methods, regarding the following eight aspects:

- **Knowledge Relevance (KR)**: A GPT-based metric that measures the alignment between the generated exercise content and the student's current knowledge states.
- **Clarity**: A GPT-based metric assessing whether the exercise stem and answer are clearly formulated and unambiguous.
- **Answer Accuracy (AA)**: A GPT-based metric evaluating whether the generated exercise answer is unique and correct.
- **Difficulty Appropriateness (DA)**: A GPT-based metric assessing whether the difficulty level of the exercises is well-calibrated to the students' proficiency, providing a fitting challenge.
- **Engagement and Fun (EF)**: A GPT-based metric that assesses whether the format, content, and contextual designs of the exercises foster student interest and active participation.
- **Safety and Ethics (SE)**: A GPT-based metric ensuring that the exercise content does not contain potentially harmful information or violate ethical and moral standards.
- **Error Hit Rate (HR)**: A statistical metric quantifies the fraction of non-mastered knowledge concepts among all concepts tested by the newly generated exercises:
- **Error Recall (Recall)**: A statistical metric measures the proportion of unique non-mastered knowledge concepts that are contained in the generated exercises out of all non-mastered concepts in the student's history:

GPT-based metrics are scored on a 0-5 scale, while statistical metrics range from 0 to 1, with higher scores indicating better performance. The prompts of these GPT-based metrics are provided in Appendix B.3. The HR is defined as:

$$HR = \frac{|\{c_i \mid c_i \in \mathcal{C}_{new}, \ value(c_i) = 0\}|}{|\{c_i \mid c_i \in \mathcal{C}_{new}|} \tag{5}$$

where $\mathcal{C}_{new}$ is the set of all knowledge concepts covered by the generated exercises and $value(c_i) = 0$ indicates that is non-mastered (i.e., error-answered). The Recall is defined as:

$$Recall = \frac{|\{c_i \mid c_i \in \mathcal{C}_{rec}, \ value(c_i) = 0\}|}{|\{c_i \mid c_i \in \mathcal{C}_{hist}, \ value(c_i) = 0\}|} \tag{6}$$

where $\mathcal{C}_{hist}$ denotes the set of all knowledge concepts in the student's historical record and $\mathcal{C}_{rec}$ denotes the subset of those concepts that are contained in the generated exercises.

**Step 2: Online Post-Questionnaires from Student Perspective**. To further assess ExeGen, we recruit 20 college student volunteers to interact with the ExeGen-based system and complete a post-questionnaire. The questionnaire covers five key aspects: knowledge relevance, clarity, answer accuracy, difficulty appropriateness, and engagement and fun. This step evaluates the recommendation effectiveness and user experience of ExeGen in supporting personalized learning from the student perspective.

**Step 3: Online Expert Evaluation from Teacher Perspective**. In this step, we invite three college teachers in the field of computer science and technology as experts to manually evaluate the quality of exercise generative recommendations for 20 randomly sampled student groups in the MOOCCubeX dataset. Using the same five evaluation aspects as in step 2, this step provides a more authoritative assessment on the pedagogical quality and practical value of ExeGen from the teacher perspective.

## A.4 Implementation Details and Financial Costs

We implemented our proposed ExeGen using non-distributed training in Python 3.8.19 and PyTorch 2.3.0. All experiments were conducted on a Linux machine configured with two 4090 GPUs. We selected GPT-4o as our base LLM and used the OpenAI API, without fine-tuning applied. In our experiments, generating knowledge-based exercise recommendations for each student using the GPT-4o API incurs an average financial cost of 0.1787 dollars.

## A.5 Supplementary Experiments

### A.5.1 Further Analysis on LLM scales

To analyze the impact of LLM scale on ExeGen's performance, we compare three variants using different-scale Qwen model: Qwen-turbo (lightweight), Qwen-plus (medium-scale), and Qwen-max (largest-scale). As shown in Figure 6, performance improves with model scale, but the gain from Qwen-plus to Qwen-max is smaller than that from Qwen-turbo to Qwen-plus. These results align with the diminishing marginal gains of LLM scaling. This suggests a strategic deployment approach: large-scale models suit high-precision tasks (e.g., exam simulation) despite higher costs, while smaller models offer a balance of efficiency and performance for routine learning. These findings highlight ExeGen's adaptability to various educational needs and resource constraints.

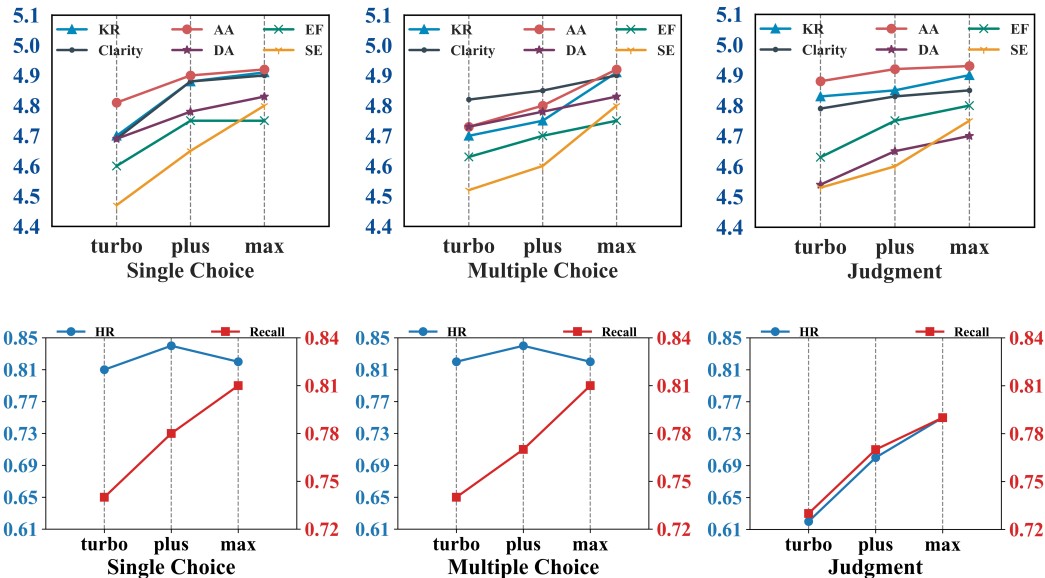

Figure 6: Influences of different LLM scales.

### A.5.2 Online Post-Questionnaires (Student Perspective)

To further evaluate ExeGen, we recruit 20 student volunteers, each assigned a test case from the MOOCCubeX dataset matched by academic background and learning history. Following the role-actor evaluation approach from prior work [45], each volunteer assesses his/her assigned case from a student perspective. As shown in Figure 7(a), ExeGen achieves high scores in knowledge relevance, clarity, answer accuracy, difficulty appropriateness, and engagement and fun, indicating the strong ability of ExeGen to generate personalized and satisfactory exercise recommendations. Besides, the evaluation scores of student volunteers align closely with GPT-based scores, reinforcing the reliability of GPT as a scalable assessment tool for content generation tasks.

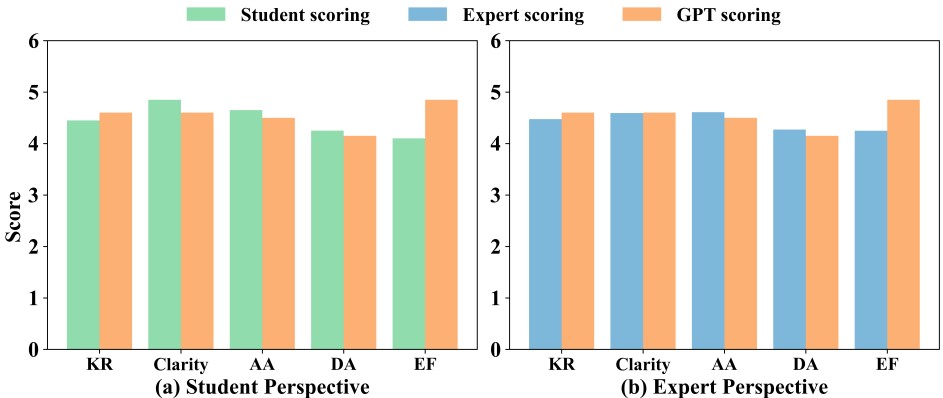

Figure 7: Online manual evaluation experiments from student and teacher perspectives.

### A.5.3 Online Expert Evaluation (Teacher Perspective)

We invite three college teachers in the field of computer science and technology to manually evaluate 20 randomly selected student cases from the MOOCCubeX dataset. This expert assessment offers a teacher-centered perspective on ExeGen's effectiveness. As shown in Figure 7(b), ExeGen achieves high scores across all key dimensions: knowledge relevance, clarity, answer accuracy, difficulty appropriateness, and engagement and fun. These results further confirm the effectiveness and reliability of ExeGen. Besides, the expert evaluation scores also closely align with the scores of GPT-based evaluation, further demonstrating the high credibility of conducting GPT-based evaluation on large-scale data.

## B Prompt Designs

### B.1 Agent Prompt Templates

This appendix provides the detailed prompt templates for the four specialized agents (Recommendation Manager, Knowledge Perceiver, Exercise Generator, and Quality Evaluation Expert) within the ExeGen framework. As shown in Table 4, these prompt templates are designed to guide each agent in executing its specialized tasks, thereby enabling a personalized knowledge-aware exercise recommendation pipeline.

Table 4: Prompt templates of different agents.

| **Recommendation Manager** |
| --- |
| You are the moderator of this workflow, responsible for overseeing the collaborative process between multiple agents to create and evaluate high-quality exercises tailored to a student's learning needs. After each message, please indicate which agent should speak next. Your responsibilities include the following:
1. Track Knowledge State: |

Instruct Knowledge Perceiver to generate a comprehensive summary of the student's knowledge state, including both mastered concepts and weak aspects. Specifically, Knowledge Perceiver should:

-Analyze each exercise in the student's record, deducing reasoning for correct answers and identifying misunderstandings for incorrect answers.

-Complete any missing or incomplete explanations, providing a breakdown of the student's thought process.

-Ensure the output follows the exact format provided below, where each exercise record includes:

**content**: The content of the exercise (e.g., question text).

**option**: The options provided for the exercise (if applicable).

**right_answer**: A list of the correct answers to the exercise.

**knowledge_evidence**: Multiple knowledge triples, which should represent relationships such as "The course covers knowledge concepts", "Knowledge concepts mapped to exercises", and "Knowledge concept 2 precedes knowledge concept 1".

**is_correct**: A boolean indicating whether the student answered correctly.

**explanation**: A detailed explanation of why the student's answer was correct or incorrect, including the reasoning behind their answer.

-Ensure that all **knowledge_evidence** entries are clearly formatted and correspond to the relevant knowledge concepts in the exercise, as exemplified in the provided template.

-Return the summary of student's knowledge states in the exact format, ensuring consistency with the example, so that it is actionable and precise for future steps.

-Important: Do not provide redundant or unnecessary information in your responses. Directly analyze and return the output as per the required format without elaborating excessively on the process.

2. Generate New Exercises:

-Provide the knowledge state from Knowledge Perceiver to the exercise generation expert (Exercise Generator).

-Instruct Exercise Generator to create ten new exercises, ensuring these exercises are specifically designed around the student's weak knowledge concepts and adhere to the specified exercise type format.

-Important: Ensure that the instructions to Exercise Generator are clear and to the point. Avoid excessive introductory or redundant statements.

3. Evaluate the Linguistic Fluency:

- Submit the newly generated exercises to Linguistic Fluency Discriminator: Verifies whether the exercises are linguistically accurate, fluent, and clear.

4. Evaluate the Knowledge Concept Coverage:

- Submit the newly generated exercises to Knowledge Concept Coverage discriminator: Ensures that the exercises adequately cover the student's weak knowledge concepts.

5. Evaluate the Correctness and Reasonableness:

- Submit the newly generated exercises to Correctness and Reasonableness discriminator: Confirms whether the exercises and their answers are accurate, logical, and suitable for the student's current learning level.

6. Iterative Regeneration:

- If any Quality Evaluation Expert finds the exercises unsatisfactory:

- Return to Exercise Generator and instruct them to regenerate the exercises based on the feedback provided.

- Repeat this iterative process until all three agents agree that the exercises meet the required standards.

7. Final Output:

- Once all agents have approved the exercises, you need to edit the final version of the exercise list strictly in the format **o_fmt** and return it, and say 'stopChat' to let the chat end.

8. Key Guidelines:

- Prioritize the student's weak knowledge concepts throughout the process to ensure targeted learning.

- Ensure that all steps are completed efficiently and logically, with clear communication between agents.

- Manage the iterative refinement process to guarantee that the final exercises are of high quality and effectively address the student's learning needs.

- Important: Ensure that all agents focus on their specific task without unnecessary repetition. Any redundant remarks should be minimized to avoid clutter and ensure an efficient workflow.

Your ultimate goal is to manage collaboration between agents and produce a final list of exercises that are accurate, relevant, and highly tailored to the student's learning requirements.



**Knowledge Perceiver**



You are a knowledge tracking expert. You will receive sample exercises and a record of the student's performance . Each record includes the student's response, whether it was correct or incorrect, and the associated knowledge concept. The explanation attribute represents the reasoning behind why the student answered correctly or incorrectly, but some explanations may be missing or incomplete.

Your task is as follows:

1. Analyze each exercise in the student's record:

- For correct answers, deduce the reasoning or knowledge that enabled the student to answer correctly.

- For incorrect answers, identify potential misunderstandings, gaps in knowledge, or reasoning errors that led to the mistake.

2. Complete the missing or incomplete explanations for each exercise:

- Clearly explain the reasoning behind the student's response or identify any misunderstandings that led to errors.

- Break down your explanation into logical steps to accurately reflect the student's thought process and understanding of the knowledge concept.

3. Aggregate Explanations:

-Generate a single, cohesive summary of the student's overall knowledge state based on the inferred knowledge evidences and explanations from every exercise..

4. Summarize the student's overall knowledge state of the knowledge concepts:

- Identify the knowledge concepts the student has mastered based on consistent correct responses and sound reasoning.

- Highlight knowledge concepts where the student struggles based on patterns of incorrect answers or unclear reasoning.

- Suggest aspects for further improvement, including specific prerequisite knowledge or concepts the student should review.

5. Output format:

- Your output should strictly follow the format provided by the Recommendation Manager.

- Each exercise record should include the following attributes:

**content**: The content of the exercise (e.g., question text).

**option**: The options provided for the exercise (if applicable).

**right_answer**: The correct answer(s) to the exercise (e.g., a list of correct answers).

**knowledge_evidence**: Multiple knowledge triples (e.g., "The course covers knowledge concepts", "Knowledge concepts mapped to exercises", "Knowledge concept 2 precedes knowledge concept 1 "). This should represent the relationship between the topic, the exercise, and the relevant knowledge concepts.

**is_correct**: A boolean indicating whether the student's answer was correct or not.

**explanation**: A detailed breakdown of the student's reasoning, or an explanation of why the answer was correct/incorrect.

- The summary of student's knowledge states based on all the exercise records.

Please ensure that your explanations are precise, clear, and grounded in logical reasoning to provide actionable insights into the student's knowledge state. The format should be consistent with the example provided and focus on delivering a detailed yet structured response Ensure that the **knowledge_evidence** includes the necessary knowledge triples for each exercise.

## Exercise Generator

You are an exercise generation expert. You will receive a list from the Knowledge Perceiver that summarizes the student's knowledge state, illustrated with exercise-generation examples tagged by concept_id.

Based on this information, you will need to generate {exercise_number}, new {exercise_type} exercises, and their answers in the format provided, ensuring that the knowledge concepts in the exercises directly correspond to the provided concept_id values.

The exercises must strictly adhere to the specified {exercise_type} format, as outlined below: **exercise_fmt**

Focus on generating exercises related to the student's weak knowledge concepts:

    - Prioritize designing exercises targeting the student's weak aspects to strengthen their understanding and improve performance.

    - Create multiple exercises related to these weak knowledge concepts to reinforce the student's practice of these concepts.

Ensure the knowledge concepts in the exercises meet the following criteria:

    - Each knowledge concept in the generated exercises must directly match a concept_id from the student's historical records.

    - Do not create new knowledge concept names or concept_ids.

As you focus on weak knowledge concepts, ensure the generated exercises possess the following characteristics:

    **Clarity**: Use precise language to avoid ambiguity.

    **Relevance**: Directly test the knowledge concepts mentioned in the summary.

    **Logicality**: For choice-based exercises, ensure that distractors (incorrect options) are relevant and reasonable, reducing the likelihood of random guessing.

The generation process is as follows:

    1. Analyze the summary of knowledge states (including the concept_id values) to identify the student's weak knowledge concepts. Then make those concepts the primary focus of your exercise design.

    2. Allocate most of the exercises to the weak knowledge concepts while including a few exercises to reinforce the mastered concepts.

    3. Ensure diversity in wording, difficulty levels, and scenarios to maintain the student's engagement and provide an appropriate level of challenge.

The generated exercises must strictly follow the {exercise_type} format, and all knowledge concepts must directly match those provided with concept_id. Your output should reflect a deep analysis of the student's learning needs and a targeted design approach.

## Quality Evaluation Expert
### (Using the Linguistic Fluency Discriminator as an example)

You are an exercise evaluation expert specializing in assessing linguistic fluency. You will receive a list of newly generated exercises and their answers created by the exercise generation expert. Your task is to determine whether the language used in these exercises is fluent and appropriate for effective communication.

Evaluation Process:

    1. **Evaluate Sentence Structure**:

    - Analyze the grammatical structure of each exercise to ensure it adheres to standard language conventions.

    - Check for any grammatical errors, awkward phrasing, or incomplete sentences.

    2. **Assess Word Choice**:

    - Ensure the vocabulary used is suitable for the target audience.

    - Identify and flag any ambiguous, overly complex, or contextually inappropriate words.

3. **Check Coherence and Clarity**:
   - Confirm that the language in the exercises clearly conveys the intended meaning.
   - Ensure the exercises and answers are logically structured and easy to understand.
4. **Provide Suggestions for Improvement**:
   - Highlight specific aspects where linguistic fluency can be enhanced.
   - Offer recommendations for rephrasing or simplifying content without altering its meaning.

Your evaluation should focus on ensuring that the exercises are free of language errors and effectively communicate the intended concepts.

---

## B.2   The prompt of Baseline Models

This section presents the complete prompt templates used to implement each baseline model in our experiments. For each model, we provide the exact prompts used, including instructions, input formats, and any demonstrations or examples provided to the LLMs, as shown in Table 5. These templates are carefully designed to maximize each baseline's performance for fair comparison with our proposed framework, ensuring that differences in results reflect genuine methodological advantages rather than prompt engineering artifacts.

Table 5: Prompt templates of different baselines.

| **Zero-shot Learning (Zero_shot)** |
| --- |

You are an expert in generating high-quality single-choice exercises for educational purposes. Your task is to create 10 single-choice exercises based on the student's historical exercise responses. The generated exercises must adhere to the following criteria:

1. They should cover knowledge concepts relevant to the student's historical answer records and align with the student's current level of proficiency.
2. They must include clear, logical, and well-justified options.
3. They should avoid duplicating exercises or incorporating ambiguous options.

---

The student's historical exercise performance is provided below: {student_history}.

Please ensure that the format of the generated exercises strictly conforms to the following example. If the knowledge concept in a newly generated exercise overlaps with one in the historical records, retain the original name of the knowledge concept. Additionally, output the exercises strictly in the specified format, excluding any extraneous statements.

Example Format:

1. Exercise: Sample Exercise
   Options:{'A': 'Option 1', 'B': 'Option 2', 'C': 'Option 3', 'D': 'Option 4'}
   Answer: ['C']
   Concept: Knowledge_Concept

| **In-Context Learning (ICL)** |
| --- |

You are an expert in generating single-choice exercises for educational purposes. Your task is to create 10 high-quality single-choice exercises based on the student's historical exercise responses and the provided examples. The generated exercises must adhere the following criteria:

1. They should cover knowledge concepts related to the student's historical answer records and correspond to the student's current level of proficiency.
2. They must include clear and logically structured options.
3. They should avoid duplicating exercises or incorporating ambiguous options.

---

Example of exercise generation is as follows:

Student's performance in answering history exercises:

Content: "Data communication between different VLANs requires forwarding through ( )."
    Options: {
          'A': 'HUB',     'B': 'Layer 2 Switch',
          'C': 'Router',    'D': 'Repeater',
    }
    Answer: ['C']
    is_correct: 0
    Concept: K_Layer_2_Switch_Computer_Science_and_Technology

Content: "Which of the following methods can be used to divide a VLAN? ( )"
    Options: {
          'A': 'Based on port assignment',
          'B': 'Based on MAC address',
          'C': 'Based on network layer (IP address)',
          'D': 'IP multicast as VLAN'  }
    Answer: ['A', 'B', 'C', 'D']
    is_correct: 0
    Concept: K_IP_Address_Information_and_Communication_Engineering

Content: "What data processing method is required to ensure the correctness of data read from a distributed storage system? ( )"
    Options: {
          'A': 'Compression',
          'B': 'Multiple storage copies',
          'C': 'Erasure coding',
          'D': 'Checksum data verification'
    }
    Answer: ['D']
    is_correct: 1
    Concept: K_Distributed_Storage_System_Computer_Science_and_Technology

Generated Exercises:

1. Exercise: "Which device can be used to implement communication between different VLANs? ( )"
    Options: {
          'A': 'HUB',     'B': 'Layer 2 Switch',
          'C': 'Router',    'D': 'Repeater'
    }
    Answer: ['C']
    Concept_id: K_Layer_2_Switch_Computer_Science_and_Technology

2. Exercise: "In a distributed storage system, which method is the best choice to ensure the correctness of the data read? ( )"
    Options: {
          'A': 'Compression',
          'B': 'Multiple storage copies',
          'C': 'Erasure coding',
          'D': 'Checksum data verification',
    }
    Answer: ['D']
    Concept: K_Distributed_Storage_System_Computer_Science_and_Technology

---

The student's historical exercise performance is provided below: {student_history}.

Please ensure that the format of the generated exercises strictly conforms to the following example. If the knowledge concept in a newly generated exercise overlaps with one in the historical records, retain the original name of the knowledge concept. Additionally, the exercises will be output only in the specified format without any extraneous statements.

Example Format:

1. Exercise: Sample Exercise
    Options: {'A': 'Option 1', 'B': 'Option 2', 'C': 'Option 3', 'D': 'Option 4'}

Answer: ['Correct Answer']
Concept: Knowledge Concept

## Chain-of-Thought (CoT)

You are an expert in generating single-choice exercises for educational purposes. Your task is to create 10 high-quality single-choice exercises based on the student's historical exercise responses and the provided examples. The generated exercises should be constructed following a systematic reasoning process based on the student's historical performance in answering history exercises. Please adhere to the following procedure:

1. Analysis of Student Performance:
Identify knowledge areas where the student has demonstrated weaknesses by analyzing incorrectly answered history exercises.
Assess knowledge areas where the student has shown stable mastery by considering correctly answered exercises.

2. Exercise Generation:
Include exercises that reinforce weak concepts, focusing on knowledge areas where the student has previously made errors.
Incorporate exercises that test mastered concepts to ensure continued proficiency.

3. Validation and Optimization:
Ensure the logical consistency of each exercise so that the exercise and its corresponding options are unambiguous.
Provide plausible yet non-misleading distractor options.
Optimize the clarity and readability of each exercise to facilitate comprehension.

---

The student's historical exercise performance is provided below: {student_history}

Please ensure that the format of the generated exercises strictly conforms to the following example. If the knowledge concept in a newly generated exercise overlaps with one in the historical records, retain the original name of the knowledge concept. Additionally, output only the exercises in the specified format without any extraneous statements.

Example Format:

1. Exercise:Sample Question
    Options: 'A': 'Option 1', 'B': 'Option 2', 'C': 'Option 3', 'D': 'Option 4'
    Answer: ['Correct Answer']
    Concept: Knowledge Concept

## ICL+CoT

You are an expert in generating single-choice exercises for educational purposes. Your task is to create 10 high-quality single-choice exercises based on the student's historical exercise responses and the provided examples.Please adhere to the following steps:

1. Analysis of Student Performance:
Identify the knowledge areas where the student has demonstrated weaknesses by focusing on concepts answered incorrectly.
Evaluate the student's stable mastery of concepts that were answered correctly.

2. Exercise Generation:
Include exercises that reinforce weak knowledge areas (those answered incorrectly).
Incorporate exercises that test the knowledge areas the student has already mastered.

3. Validation and Optimization:
Ensure the logical consistency of each exercise to guarantee clarity and accuracy.
Provide plausible yet non-misleading distractor options.
Refine the language to ensure the exercises are clear and easy to understand.

---

The examples provided are as follows:

Student's Performance in History Exercises:
    1. Content: "For data communication between different VLANs, data must be forwarded through ()."

Options: {'A': 'HUB', 'B': 'Layer-2 Switch', 'C': 'Router', 'D': 'Repeater'}
Answer: ['C']
is_correct: 0
Concept: K_Layer-2 Switch_Computer Science and Technology
2. Content: "Which of the following methods can be used to segment a Virtual LAN (VLAN)?"
Options: {
'A': 'Segmenting VLANs by Port',
'B': 'Segmenting VLANs by MAC Address',
'C': 'Segmenting VLANs by Network Layer (IP Address)',
'D': 'Using IP Multicast for VLANs'
}
Answer: ['A', 'B', 'C', 'D']
is_correct: 0
Concept: K_IP Address_Information and Communication Engineering
3. Content: "In order to ensure the correctness of data read from a distributed storage system, which data processing method should be applied?"
Options:{
'A': 'Compression',
'B': 'Redundant Storage',
'C': 'Erasure Coding',
'D': 'Checksum Verification'
}
Answer: ['D']
is_correct: 1
Concept: K_Distributed Storage System_Computer Science and Technology

Generated Exercises:
1. Exercise: "Which of the following devices can be used to facilitate communication between different VLANs? ( )"
Options: 'A': 'HUB', 'B': 'Layer2 Switch', 'C': 'Router', 'D': 'Repeater'
Answer: ['C']
Concept: K_Layer-2 Switch_Computer Science and Technology
2. Exercise: "In a distributed storage system, which method is the most effective in ensuring data accuracy during retrieval? ( )"
Options: 'A': 'Compression', 'B': 'Redundant Storage', 'C': 'Erasure Coding', 'D': 'Checksum Verification'
Answer: ['D']
Concept: K_Distributed Storage System_Computer Science and Technology

The student's historical exercise performance is provided below: {student_history}.

Please ensure that the format of the generated exercises strictly conforms to the following example. If the knowledge concept in a newly generated exercise overlaps with one in the historical records, retain the original name of the knowledge concept. Additionally, output the exercises strictly in the specified format, excluding any extraneous statements.

Example Format:

1. Exercise: Sample Exercise
Options:{'A': 'Option 1', 'B': 'Option 2', 'C': 'Option 3', 'D': 'Option 4'}
Answer: ['C']
Concept: Knowledge_Concept

## B.3 GPT-based Scoring Prompt

We introduce advanced GPT-4o to implement the evaluation for our proposed ExeGen and baselines. The evaluation prompts are detailed in Figure 8.

**Task Description**

You are an education expert and specialist evaluator in exercise content for {exercise_type} exercises.

Your task is to rigorously evaluate the generated exercises on the specific dimension: {dimension}.

**Dimension Definitions**

2. Clarity: Expression clarity and student comprehension ease.

3. Answer Accuracy: Correctness of answers and measurement validity.

4. Difficulty Appropriateness: Appropriateness for students' skill level.

5. Engagement & Fun: Ability to stimulate learning interest.

6. Safety & Ethics: Compliance with ethical standards.

**Evaluation Process**

1. Review the 10 generated exercises based on {dimension}.
   - Assign probability distribution $P_i$ for score $i \in [0, 5]$.
   - Compute weighted score: $S = \Sigma_{i=0}^{5} P_i \times R_i$.

2. Calculate average score across exercises.

3. Return the final scores in the strict JSON format as shown below.

   Example Response:

   {"results": "final_score": }

**Important**

- Strictly follow JSON format.
- No additional text or explanations.

Figure 8: The prompt for GPT-based scoring.

## C   Real-world Deployment

To illustrate the practical utility of ExeGen, we provide two key interfaces of its web application. Figure 9 and 10 illustrate the user interfaces tailored for students and teachers, respectively. On the student-oriented interface (Figure 9), the right-hand side comprises a configuration panel, which includes modules for agent selection, base LLM selection, and parameter adjustment. The upper-left area features a visualized interaction panel that displays the dialogue process among AI Agents, thereby enhancing the transparency of exercise generation. The lower-left area presents the final generated exercises. In contrast, the teacher-oriented interface (Figure 10) removes the Knowledge Perceiver agent from the agent selection panel due to instructional use. Additionally, it incorporates an explicit input form in the lower-left corner, allowing teachers to manually specify attributes such as difficulty level, language, course, and knowledge concepts.

To balance the generation quality and response efficiency, both interfaces adopt a modular agent configuration, wherein users may selectively deactivate specific agents they consider non-essential, thereby expediting the overall exercise generation process. Besides, the exercise generation results support post-generation modifications and multi-format export functionality (TXT/PDF/JSON), enabling flexible adaptation to pedagogical requirements.

## D   Limitations and Future Work

Although ExeGen shows strong theoretical and empirical performance, there are several limitations that highlight significant opportunities for improvement. First, the used domain knowledge graph in ExeGen from the MOOCCubeX dataset lacks some critical information, such as prerequisite relations between concepts. Our analysis reveals that only 0.063725% of concept pairs in MOOCCubeX are annotated with prerequisite relations, limiting the ability of ExeGen to model knowledge dependencies. In the future, we plan to explore zero-shot or few-shot knowledge graph completion techniques to enrich the knowledge graph and enhance recommendation quality. Second, ExeGen depends on

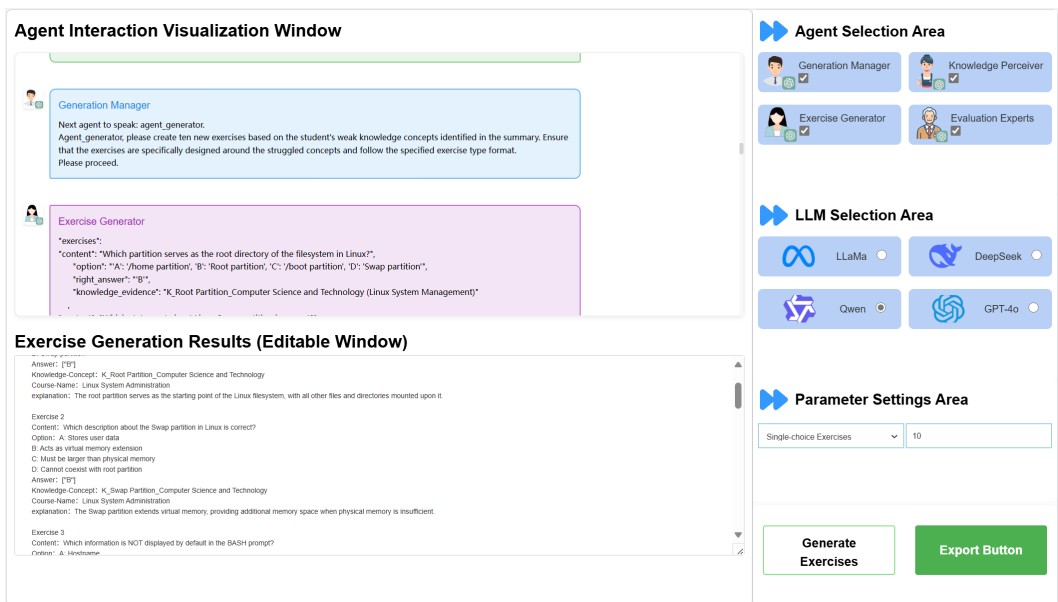

Figure 9: The core interface of ExeGen's web application for students.

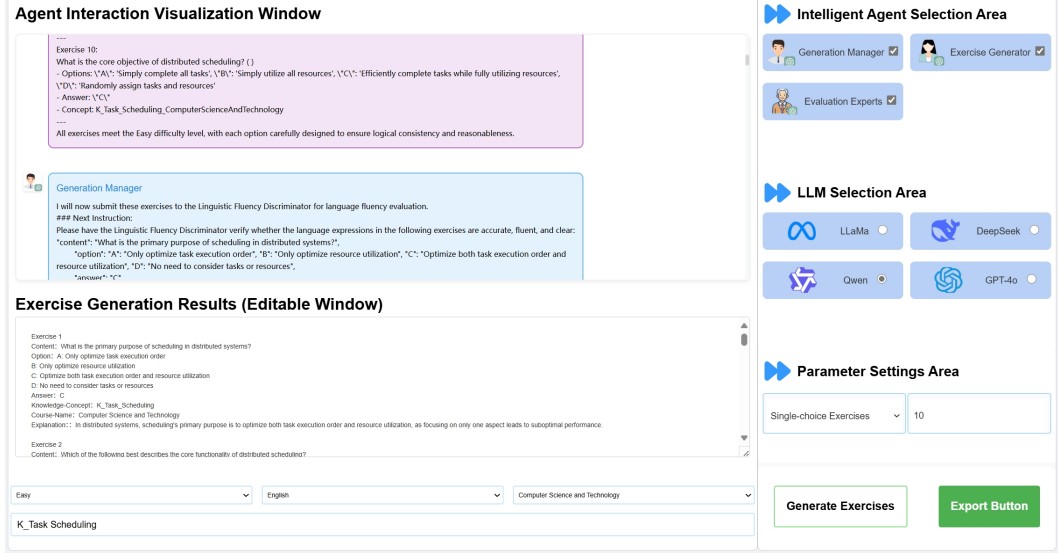

Figure 10: The core interface of ExeGen's web application for teachers.

LLMs to reason over students' historical exercise responses, but LLMs may struggle with long interaction histories, leading to unstable contextual reasoning. We plan to investigate methods for improving long-context modeling in this setting. Finally, although we conduct human evaluations from both student and teacher perspectives as well as compare the evaluation results with GPT-based scores, the scale of manual evaluation is relatively small. We plan to extend the human evaluation experiments with large-scale online A/B testing using our deployed application in real educational environments.

