# OpenReview forum: "Knowledge Starts with Practice: Knowledge-Aware Exercise Generative Recommendation with Adaptive Multi-Agent Cooperation"
_NeurIPS.cc/2025/Conference — NeurIPS 2025 poster_

### Official Review · Reviewer_G5ta · 2025-06-27

**Clarity:** 3
**Significance:** 4
**Originality:** 3
**Rating:** 5
**Confidence:** 5

**Summary:**

This paper presents ExeGen, a novel multi-agent framework for Knowledge-aware Exercise Generative Recommendation (KEGR). By coordinating LLM-powered agents for knowledge perception, exercise generation, and quality refinement, ExeGen provides personalized and interpretable learning support. Extensive experiments and real-world deployment validate its effectiveness in adaptive education.

**Questions:**

1. In this work, all exercises are generated by LLMs. Although a corresponding evaluation expert system is designed, it still cannot fundamentally address the issue of ensuring that the generated exercises fully comply with educational standards, especially for complex problems.
2. In addition to methods like CoT, I believe it would be beneficial to incorporate approaches such as Self-Consistency [1] and Best-of-N [2] to further validate the superiority of the proposed method.
3. The current framework only supports objective exercise types (single choice, multiple choice, and judgment), and has not yet addressed other types of exercise.

[1]Self-consistency improves chain of thought reasoning in language models. ICLR 2023.

[2]Let’s verify step by step. ICLR 2024.

**Ethical Concerns:**

["NO or VERY MINOR ethics concerns only"]

**Final Justification:**

Thanks author's response. I will maintain my original positive score.

**Limitations:**

yes

**Quality:**

3

**Strengths And Weaknesses:**

Strengths:
1. The proposed KEGR framework dynamically infers students’ knowledge states from their past exercise responses and customizes the generation of new exercises, making the design more aligned with real-world learning scenarios.
2. The incorporation of a human-simulated knowledge perception mechanism and an adversarial refinement loop ensures both the interpretability of the learner's knowledge state and the high quality of the generated exercises.
3. The proposed method has been deployed in real school settings, with application screenshots provided, which strongly demonstrates the practical validity of the approach.

Weaknesses:
1. To the best of my knowledge, even the most powerful large language models still suffer from hallucination issues. Since all the exercises in the paper are generated by LLMs, the proposed evaluation expert system—though carefully designed—cannot fundamentally guarantee that the generated exercises fully comply with educational standards.
2. Due to the lack of directly comparable baselines, the authors adopt four representative methods. It may be worth considering the inclusion of some advanced LLM reasoning variants (e.g., Tree-of-Thoughts) to further demonstrate the effectiveness of the proposed approach.
3. The framework currently supports only objective exercise types (single choice, multiple choice, judgment), omitting more complex subjective or open-ended tasks that are critical in higher-order learning assessment.

---

> ### Author Rebuttal · Authors · 2025-07-31
>
> **Response to W1&Q1:**
>
> Thank you for your insightful comments. We fully agree that hallucination remains a key limitation of LLMs particularly for educational applications. We would like to clarify our design motivation and hallucination mitigation strategies.
>
> First, we build our ExeGen framework on LLMs such as GPT due to their strong reasoning capabilities, cross-disciplinary versatility, and language generation diversity. These strengths enable ExeGen to infer students’ knowledge states from historical responses and generate tailored exercises at scale. Recent studies [1,2] have shown that, when paired with carefully designed prompts and multi-step reasoning, LLMs can be successfully applied to education scenarios, such as personalized tutoring. Therefore, leveraging ExeGen for personalized exercise generation is feasible and important to achieve large-scale personalized teaching for real-world applications.
>
> Second, we recognize that generating high-quality, knowledge-aware, and personalized exercises remains a non-trivial task. Simple prompting often leads to shallow, unstructured, or inaccurate content. To address this, we introduce several novel designs in our proposed ExeGen, including a multi-agent cooperation framework, a human-simulated knowledge perception mechanism, and an exercise generative-adversarial mechanism, to improve reasoning coherence and educational alignment.
>
> Third, to eliminate potential hallucinations, we incorporate three technical innovations and one practical safeguard: 1) A human-simulated knowledge perception mechanism enforces step-by-step reasoning through a structured process of “knowledge retrieval→analysis→exercise solving”, improving transparency and reducing single-step generation errors. 2) We embed factual knowledge from domain-specific knowledge graphs into prompts and apply a dual-matching strategy to select relevant high-quality demonstrations, guiding generation toward pedagogically accurate and context-aware outputs. 3) An exercise generative-adversarial mechanism introduces expert agents to evaluate and refine exercises along three dimensions (linguistic fluency, coverage of erroneous knowledge concepts, and the correctness of exercises) through adversarial feedback. 4) As shown in the application interface screenshots in Appendix C, the generated exercises are editable in the application interface, allowing teachers to manually review and adjust final outputs before posting to students.
>
> In summary, ExeGen combines LLM strengths with structured control and teacher-in-the-loop validation to reduce hallucination risks while enabling practical deployment. Notably, ExeGen is designed as an AI-assisted tool to support teachers rather than replace them. In this way, teachers shift from exercise creators to reviewers, enabling human-in-the-loop validation that aligns generated content with instructional goals. We appreciate your feedback and will explore advanced hallucination mitigation strategies such as reinforcement learning with human feedback and prompt tuning in future work.
>
> [1] SocraticLM: Exploring Socratic Personalized Teaching with Large Language Models, NeurIPS 2024.
>
> [2] What Makes In-context Learning Effective for Mathematical Reasoning, ICML 2025.
>
> **Response to W2&Q2:**
>
> Thank you for the constructive suggestions. To strengthen the comparison, we add three advanced LLM reasoning variants as additional baselines, including Tree-of-Thoughts (ToT), Self-Consistency (SC), and Best-of-N (BoN). Due to time and space constraints, we evaluate these methods on the single-choice exercise setting. As shown in the following table, these advanced baselines slightly outperform the original CoT baseline. And our proposed ExeGen framework consistently outperforms all baselines across all metrics. These results further confirm the robustness and effectiveness of ExeGen.
> |Methods|KR|Clarity|AA|DA|EF|SE|HR|Recall|
> |:-:|:-:|:-:|:-:|:-:|:-:|:-:|:-:|:-:|
> |ToT|4.75|4.70|4.82|4.75|4.68|4.60|0.65|0.70|
> |SC|4.55|4.50|4.60|4.55|4.45|4.50|0.58|0.62|
> |BoN|4.60|4.55|4.65|4.60|4.50|4.55|0.60|0.65|
> |ExeGen|4.81|4.78|4.89|4.81|4.72|4.62|0.70|0.75|
>
> We appreciate your suggestions, which help us further validate ExeGen’s superiority over recent LLM reasoning techniques.
>
> **Response to W3&Q3:**
>
> Thank you for your insightful comments. We fully agree that subjective and open-ended exercises play a significant role in higher-order learning assessment. While the current paper focuses on objective exercise types with clear structure and evaluation criteria to ensure experimental rigor and reproducibility, we emphasize that the proposed ExeGen framework is inherently adaptable to generating subjective and open-ended exercises, such as code reading and completion.
>
> We chose not to include these exercise types in the current paper for the following reasons: 1) Such exercises often lack unique answers and objective evaluation metrics, making automated evaluation difficult and less reliable; 2) Human evaluation for subjective or open-ended exercises requires domain expertise and is time-consuming, which limits scalability; 3) These exercises involve more complex text generation, increasing the risk of hallucination and higher token consumption. For these reasons, we focus on the objective exercise generation task that is more controllable and has quantifiable evaluation criteria in this paper.
>
> We take your comments very seriously and are actively exploring the design of scalable human-computer collaborative evaluation mechanisms for subjective and open-ended exercise generation tasks. We would greatly appreciate the opportunity to further discuss the generation and assessment mechanisms for such exercises with you. Your professional expertise will provide valuable insights for our research. Once again, thank you for your attention and support.

---

> > ### Comment · Area_Chair_uT2w · 2025-08-05
> >
> > The authors have provided a rebuttal to your comments, and it's an important part of the review process to give their response careful consideration. Please take a moment to review their rebuttal and provide any follow-up comments. This will help ensure there’s sufficient time for discussion and any necessary follow-up.
> >
> > Best regards,
> >
> > AC

---

> ### Author Response · Authors · 2025-08-07
>
> Dear reviewer G5ta,
>
> Thank you sincerely for your thoughtful and constructive review of our paper. We are grateful for your positive recognition of the innovation, practicality, and actual deployment effect.
>
> First, we greatly appreciate your valuable suggestions regarding potential limitations and directions for improvement. In particular, in response to your suggestion on incorporating more advanced LLM reasoning baselines, we have conducted additional comparison experiments including Tree-of-Thoughts, Self-Consistency, and Best-of-N. These new baselines slightly outperform CoT, but our proposed ExeGen still consistently outperforms all of them across all metrics, further validating its robustness and superiority. The results are included in our updated rebuttal.
>
> Second, we also fully agree with your comment regarding the importance of subjective and open-ended exercise types in higher-order learning. As mentioned in our rebuttal, while this paper focuses on objective exercise generation tasks to ensure rigor and evaluation consistency, the core design of ExeGen is flexible and extensible to subjective exercise generation, which we are actively exploring in ongoing work.
>
> Lastly, we deeply value your feedback on hallucination risks in LLM-based generation. While we have integrated multiple technical modules and human-in-the-loop design for mitigation, we believe this is a significant long-term challenge. We would continue to improve along this direction, including future exploration of reinforcement learning with human feedback and advanced prompt tuning techniques.
>
> As the rebuttal phase deadline is approaching, we kindly invite you to review our updated response. If our clarifications and additional results have sufficiently addressed your concerns, we would be sincerely grateful if you could consider raising your score. Your support is very important to us and this research.
>
> Thank you again for your time and your generous support of our research.
>
> Best regards,
>
> Authors

---

> ### Author Response · Authors · 2025-08-09
>
> Dear Reviewer G5ta,
>
> We are very grateful for your encouraging review and recognition of both the novelty and practical value of our work. Your constructive suggestions on extending our framework to more complex exercise types have provided us with valuable guidance for our future research. We sincerely appreciate your patience, time, and effort.
>
> Best regards,
>
> Authors

---

### Official Review · Reviewer_7wBG · 2025-07-03

**Clarity:** 3
**Significance:** 3
**Originality:** 3
**Rating:** 4
**Confidence:** 4

**Summary:**

This paper proposes a novel and practical task _Knowledge-aware Exercise Generative Recommendation_, and introduces a LLM-based multi-agent framework. Its thoughtful design and real-world validation demonstrate strong potential for adaptive learning. The real-world deployment supports the claims of effectiveness and superiority.

**Questions:**

1. The manuscript acknowledges that LLMs may hallucinate. To address this, it introduces a human-simulated knowledge perception mechanism based on a domain knowledge graph, involving three steps: relevant knowledge retrieval, analysis, exercise solving. It is recommended to evaluate this pipeline experimentally and identify subject areas where it may not perform well.
2. The experimental setup lacks clarity, especially regarding the subject domains (e.g., math, science). Since knowledge structures vary across subjects, it's important to test the method’s effectiveness in different domains and discuss whether the pipeline needs subject-specific adaptations.
3. The completeness of the knowledge graph is crucial for accurate knowledge state estimation. A more thorough analysis of its coverage (especially missing prerequisite links), is needed to assess its impact on system performance.

**Ethical Concerns:**

["NO or VERY MINOR ethics concerns only"]

**Final Justification:**

1. The authors performed new tests to address the concerns about the impact of incomplete knowledge graphs.
2. The authors clarified that the knowledge graph acts as auxiliary guidance and the model is not fundamentally dependent on its completeness.
3. The rebuttal reaffirmed that the main contributions of the paper, such as the KEGR task, the multi-agent framework, and the enhancement mechanisms, are robust and effective regardless of the knowledge graph's completeness.

**Limitations:**

yes

**Quality:**

3

**Strengths And Weaknesses:**

Strength:
1. Bridges knowledge tracing and exercise recommendation, forming a closed-loop adaptive system.
2. Four LLM agents (Manager, Perceiver, Generator, Evaluator) work in a feedback loop, effectively handling the KEGR task.
3. The real-world deployment demonstrates its effectiveness.

Weakness:
1. While the proposed human-simulated knowledge perception mechanism aims to reduce LLM hallucination through a three-step process (knowledge retrieval, analysis, exercise solving), its effectiveness has not been independently validated.
2. The current experiments focus on a single subject domain, but this is not clearly stated. Since knowledge structures differ significantly between disciplines, it remains unclear how well the proposed method generalizes across domains or whether subject-specific adjustments are needed.
3. The domain knowledge graph used in this study lacks completeness, particularly in prerequisite relationships between concepts. This limitation may affect the accuracy of knowledge state modeling and the quality of generated exercises. A more detailed analysis of the knowledge graph’s coverage and its impact on system performance is needed.

---

> ### Author Rebuttal · Authors · 2025-07-31
>
> **Response to W1&Q1:**
>
> Thank you for the recognition of our work and the thoughtful comments. We agree that validating the effectiveness of the human-simulated knowledge perception mechanism is essential.  This mechanism aims to emulate human exercise-solving processes: knowledge retrieval→analysis→exercise answering, to accurately perceive the personalized knowledge states of students. In this paper, we have conducted multiple experimental evidence to support its effectiveness:
>
> **Ablation study** (§5.3, Table 2): Removing the human-simulated knowledge perception mechanism (w/o. HKP) results in a significant performance drop, confirming its critical role and effectiveness for knowledge state perception and personalized exercise generation.
>
> **Human expert evaluation** (Appendix§A.5.3, Figure 7): Teachers rate the generated knowledge state descriptions with an average Knowledge Relevance (KR) score of 4.47 out of 5, indicating strong alignment with students’ actual response performance.
>
> **GPT-based scoring** (§5.3, Table 1): The KR score rated by GPT-4o averages 4.60, further supporting the consistency between generated knowledge states and exercises.
>
> **Case studies** (§5.5, Figure 5): A real case illustrates that ExeGen accurately identifies students’ underlying conceptual weakness from their historical responses.
>
> Additionally, we conduct a new experiment focusing specifically on the generated knowledge state descriptions. Results across traditional metrics (Recall, Language Fluency) and GPT-based scoring metrics (Clarity, Safety and Ethics, Consistency, and Granularity) confirm the precision and reliability of the knowledge state modeling.
> |Methods|Recall$\uparrow$|Language Fluency$\downarrow$|Clarity$\uparrow$|Safety and Ethics$\uparrow$|Consistency$\uparrow$| Granularity$\uparrow$|
> |:-:|:-:|:-:|:-:|:-:|:-:|:-:|
> |ExeGen|0.84|45.18|4.31|4.47|4.37|4.30|
>
> These results offer quantitative evidence of the mechanism’s effectiveness. We appreciate your comment, which helps us strengthen the effectiveness demonstration of our work.
>
> **Response to W2&Q2:**
>
> Thank you for raising this important point. We clarify that although our main experiments focus on the computer science domain, the core components of ExeGen, including the multi-agent cooperation framework, human-simulated knowledge perception mechanism, and exercise generative-adversarial mechanism, are designed to be domain-agnostic. ExeGen relies on three transferable elements: 1) domain-specific knowledge graphs; 2) human-cognition-inspired multi-step reasoning guided by demonstrations; and 3) iterative multi-dimensional quality optimization. These elements can be adapted to other subjects with well-defined knowledge structures, such as mathematics or physics. To assess generalization, we conduct additional experiments on the mathematics domain using the MOOCCubeX dataset. Due to time and space constraints, we only evaluate on single-choice exercise generation. As shown in the table below, ExeGen consistently outperforms all baselines across all metrics, demonstrating its strong generalization.
> |Methods|KR|Clarity|AA|DA|EF|SE|HR|Recall|
> |:-:|:-:|:-:|:-:|:-:|:-:|:-:|:-:|:-:|
> |Zero_shot|3.68|3.53|3.6|3.38|3.12|4.12|0.11|0.18|
> |ICL|4.02|4.07|4.01|3.92|3.95|4.05|0.32|0.41|
> |CoT|4.15|4.18|4.06|4.00|3.80|4.00|0.30|0.39|
> |ICL+CoT|4.30|4.33|4.31|4.25|4.15|4.30|0.22|0.49|
> |ExeGen|4.72|4.62|4.65|4.70|4.50|4.85|0.62|0.70|
>
> Moreover, we conduct additional experiments on MathDial, an English dialogue-based math dataset with 2,848 test samples. The experimental table below shows that ExeGen consistently outperforms all baselines across all metrics, further demonstrating its strong generalization ability.
> |Methods|KR|Clarity|AA|DA|EF|SE|HR|Recall|
> |:-:|:-:|:-:|:-:|:-:|:-:|:-:|:-:|:-:|
> |Zero_shot|3.60|3.45|3.50|3.30|3.00|4.20|0.08|0.15|
> |ICL|4.05|4.15|4.00|4.00|3.90|4.10|0.30|0.40|
> |CoT|4.10|4.20|4.00|3.95|3.75|3.90|0.28|0.38|
> |ICL+CoT|4.20|4.30|4.25|4.15|4.10|4.25|0.20|0.45|
> |ExeGen|4.65|4.50|4.55|4.60|4.35|4.80|0.60|0.65|
>
> **Response to W3&Q3:**
>
> Thank you for your insightful suggestions. We agree that the completeness of the domain-specific knowledge graph, particularly with regard to prerequisite relationships among concepts, might affect the quality of knowledge state perception and exercise generation. However, in ExeGen, the knowledge graph plays an auxiliary supporting role, serving as factual knowledge guidance to help the human-simulated knowledge perception mechanism mitigate the hallucinations of LLMs and improve inference reliability. It is not the core component for knowledge state modeling or exercise generation.
>
> Particularly, we appreciate your valuable opinion, which is highly relevant to our ongoing work on LLM-assisted knowledge graph completion. We hypothesize that LLMs can compensate for incomplete graphs by leveraging internalized domain knowledge. To explore this, we conduct a preliminary study on MOOCCubeX by randomly masking a subset of prerequisite triples and prompting GPT-3.5-turbo to infer the masked relations. As shown below, GPT-3.5-turbo significantly outperforms traditional knowledge graph completion methods, suggesting strong potential for improving knowledge graph coverage.
> |Metrics|TransE|DistMult|ComplEx|CP|InGram|GPT-3.5-turbo|
> |:-:|:-:|:-:|:-:|:-:|:-:|:-:|
> |ACC|29.38%|42.76%|47.58|42.10%|52.62%|62.37%|
> |F1|32.35%|49.71%|53.22%|56.72%|49.03%|63.63%|
>
> We are integrating this insight into future versions of ExeGen by adding a dedicated knowledge graph completion and refinement module. Your feedback not only confirms the value of our current design but also points to an exciting direction for further enhancement.

---

> > ### Comment · Reviewer_7wBG · 2025-08-04
> >
> > Thank you for the authors' clarification.
> >
> > The authors' response to W3&Q3 is not fully aligned with my original concern. While they demonstrate that "GPT-3.5-turbo significantly outperforms traditional knowledge graph completion methods," my primary issue was the influence of an incomplete knowledge graph on their proposed method, not a general comparison of LLMs to traditional KG completion techniques. Consequently, I feel my core concern has not been addressed.

---

> > > ### Author Response · Authors · 2025-08-05
> > >
> > > We sincerely appreciate your thoughtful comments. To further address your core concern regarding the influence of an incomplete knowledge graph, we have introduced an explicit knowledge graph completion module into the human-simulated knowledge perception mechanism of ExeGen. This module prompts the LLM to infer and supplement missing prerequisite and successor relations for concepts relevant to the given exercises. We denote this enhanced version as “w. KGC” and have conducted additional experiments, focusing on single-choice exercises due to time constraints.
> > > |Methods|KR|Clarity|AA|DA|EF|SE|HR|Recall|
> > > |:-:|:-:|:-:|:-:|:-:|:-:|:-:|:-:|:-:|
> > > |w.KGC|4.84|4.76|4.87|4.78|4.73|4.59|0.75|0.84|
> > > |ExeGen|4.81|4.78|4.89|4.81|4.72|4.62|0.70|0.75|
> > >
> > > The experimental results show that the introduction of knowledge graph completion module does indeed help improve the performance of exercise generation in some metrics, but the improvement is slight. This confirms that ExeGen does not heavily rely on the completeness of the knowledge graph. Instead, the knowledge graph serves as auxiliary support to guide factual reasoning and reduce hallucination. These findings also reinforce our earlier hypothesis in **Response to W3&Q3:**, which LLMs can implicitly complete missing relations by leveraging their internalized world knowledge. We appreciate your insightful feedback, which helped us further demonstrate and refine the design of our framework.

---

> > > ### Author Response · Authors · 2025-08-07
> > >
> > > Dear reviewer 7wBG,
> > >
> > > Thank you again for your thoughtful and constructive feedback on our paper. We deeply appreciate the time and effort you devoted to providing valuable suggestions, particularly regarding the influence of incomplete knowledge graphs on our proposed ExeGen.
> > >
> > > Following your latest comment, we have conducted additional experiments and integrated an explicit knowledge graph completion module into our framework to directly address your core concern. These new results, along with a detailed explanation, have been carefully documented in our supplemental rebuttal. These findings demonstrate that while the KGC module yields slight improvements in some metrics, the overall performance of our method remains strong even without it. This supports our view that **the knowledge graph serves primarily as auxiliary guidance**, and our **ExeGen is not fundamentally dependent on its completeness**.
> > >
> > > We sincerely hope this additional clarification demonstrates that **the core innovation and technical contributions of our work, including the formulation of the KEGR task, the design of the multi-agent framework ExeGen, and the introduction of two enhancement mechanisms, remain intact and effective** regardless of the completeness of knowledge graphs. These contributions constitute the main focus and novelty of our work.
> > >
> > > As the rebuttal phase deadline is approaching, we kindly invite you to review our updated response. If our clarifications and additional results have sufficiently addressed your concerns, we would be sincerely grateful if you could consider raising your score. Your support would be deeply appreciated, as it would help recognize our contributions to the field. It is very important for us and this research.
> > >
> > > Thank you again for your professional comments and for helping us improve the quality of this submission.
> > >
> > > Best regards,
> > >
> > > Authors

---

> ### Author Response · Authors · 2025-08-09
>
> Dear Reviewer 7wBG,
>
> Thank you for your thoughtful review and for acknowledging the contributions and real-world value of our work. Your suggestions on knowledge graph completeness have been very insightful and have inspired us to explore knowledge graph completion as a promising direction for further improvements. We sincerely appreciate your patience, time, and effort.
>
> Best regards,
>
> Authors

---

### Official Review · Reviewer_zZz2 · 2025-07-03

**Clarity:** 3
**Significance:** 2
**Originality:** 2
**Rating:** 4
**Confidence:** 3

**Summary:**

This paper proposes Knowledge-aware Exercise Generative Recommendation (KEGR), a task that unifies knowledge state diagnosis and personalized exercise generation. To achieve this, the paper develops ExeGen, a multi-agent framework powered by LLMs, where four specialized agents collaborate in a closed-loop pipeline: Recommendation Manager, Knowledge Perceiver, Exercise Generator, and Quality Evaluation Experts. Furthermore, key techniques also include a human-like knowledge perception mechanism using knowledge graphs and in-context learning, and an adversarial refinement process to improve exercise quality. Experiments on real-world datasets and deployment in college settings show ExeGen’s superiority over existing baselines, validating its potential for adaptive and interpretable learning support.

**Questions:**

1. Could the paper clarify what specific architectural or procedural innovations distinguish their multi-agent framework from simply coordinating ICL and CoT within standard LLM pipelines? What novel agent-specific interactions or functionalities are introduced beyond existing prompting paradigms?

2. Have the paper conducted or considered any longitudinal studies to measure actual student learning gains after engaging with the generated exercises? If not, how do they justify the use of GPT-based and human scoring as sufficient proxies for real educational effectiveness?

3. Given that the Knowledge Perceiver is designed to model student knowledge states, why is there no quantitative comparison with established knowledge tracing models such as BKT, DKT, or GKT?

4. Many existing frameworks like ReAct and Reflexion have demonstrated strong performance in agent-based reasoning tasks. Could the authors provide empirical comparisons with these frameworks or explain in detail why such approaches are unsuitable for the KEGR task?

5. How exactly are the Human-simulated Knowledge Perception Mechanism and the Exercise Generative-Adversarial Mechanism integrated into the four-agent architecture? Which agent(s) are responsible for executing these mechanisms, and how are they invoked in the workflow?

**Ethical Concerns:**

["NO or VERY MINOR ethics concerns only"]

**Final Justification:**

The concerns are addressed, and the score is raised. Thanks for the efforts.

**Limitations:**

yes

**Quality:**

3

**Strengths And Weaknesses:**

Strengths:

1. The paper introduces KEGR, a meaningful task that integrates knowledge tracing and exercise recommendation into a unified, closed-loop adaptive learning system.

2. ExeGen's use of four specialized LLM-powered agents—each responsible for a distinct subtask—ensures transparency and modularity in handling complex reasoning and generation tasks. Furthermore, Human-Simulated Knowledge Perception and Adversarial Quality Refinement strengthen the above agent’s ability, contributing to a more robust workflows.

3. The system is evaluated comprehensively through a hybrid approach that includes GPT-based metrics, statistical scores, student questionnaires, and teacher expert assessments, which is not only tested offline but also deployed in a real college environment, providing both quantitative and qualitative validation and also supporting its practical utility and readiness for real-world application.

Weakness:

1. The proposed agent-based framework lacks substantial technical novelty. It heavily relies on existing techniques such as In-Context Learning (ICL) and Chain-of-Thought (CoT), and does not demonstrate significant innovation in the design of the agent architecture itself.

2. The current evaluation focuses primarily on the quality of the generated exercises, assessed mainly through GPT-based scoring and human judgments. However, these metrics are inherently subjective and do not reflect the core objective of exercise recommendation—namely, the actual improvement in students' knowledge and learning outcomes as measured by their performance after doing exercised generated by this framework.

3. While the motivation of the agent design is to address limitations in both knowledge tracing and exercise recommendation, the evaluation fails to provide a detailed comparison against traditional models from these domains. For instance, the Knowledge Perceiver module is intended to trace student knowledge states, yet its performance is not compared against standard knowledge tracing models, making it difficult to assess its effectiveness.

4. Although the paper is the first to propose the KEGR task and solve it via an agent-based approach, there are already many well-established agent frameworks such as ReAct, Reflexion, etc. Including empirical comparisons with these frameworks, or clearly explaining why they are not suitable for KEGR, would significantly strengthen the contribution.

5. The paper’s presentation is somewhat confusing. For example, in addition to the four core agent’s components, the authors introduce two mechanisms—Human-simulated Knowledge Perception and Exercise Generative-Adversarial mechanisms—but do not clearly specify how these techniques are integrated into which component of the framework. Clarifying their roles and placement within the overall framework would enhance clarity.

---

> ### Author Rebuttal · Authors · 2025-07-31
>
> **Response to W1&Q1:**
>
> Thank you for the thoughtful feedback. We respectfully clarify that ExeGen is not a simple combination of existing techniques like ICL or CoT, but a task-driven, mechanism-enhanced multi-agent framework designed for a novel educational application task. Our technical contributions span task reasoning mechanisms, task formulation, and framework design.
>
> For reasoning mechanisms, we introduce two innovations that go beyond standard ICL or CoT. First, the human-simulated knowledge perception mechanism guides LLMs to emulate fine-grained cognitive reasoning over student historical responses. It incorporates domain-specific knowledge graphs, dual-matching retrieval strategies, and multi-step human-simulated recognition reasoning to produce accurate knowledge state descriptions, which surpasses standard ICL’s static prompt-based strategy and CoT’s single-chain reasoning. Second, the exercise generative-adversarial mechanism creates an iterative feedback loop between the exercise generator and evaluation expert agents, enabling multi-dimensional educational critique and refinement, which improves generation beyond one-pass Cot/ICL.
>
> For task formulation, we define KEGR as a new task that reformulates and unifies Knowledge Tracing (KT) and Exercise Recommendation (ER) into a generation-based setting. Unlike these binary classification (KT) or matching (ER) tasks, KEGR aims to dynamically perceive students’ knowledge states from historical exercise responses, and subsequently generate tailored exercises based on the inferred knowledge states.
>
> For framework design, ExeGen is a modular architecture with four collaborative agents, responsible for global supervision, knowledge state perception, exercise generation, and quality refinement, respectively. This setup supports complex-task-oriented adaptive reasoning, interpretability, and future extensibility, beyond the combination of ICL and CoT.
>
> Finally, as shown in Table 2 (§5.3), ExeGen significantly outperforms strong baselines including CoT, ICL, and hybrid methods, confirming its improved reasoning and generation capabilities.
>
> In summary, ExeGen contributes new technical mechanisms and a task-specific multi-agent architecture that meaningfully extend LLM reasoning for personalized education.
>
> **Response to W2&Q2:**
>
> Thank you for your insightful comment. We fully agree that the ultimate goal of educational AI systems is to improve student learning outcomes. Our current work focuses on a key prerequisite: generating high-quality, personalized exercises based on students’ historical responses. This forms the foundation for long-term learning gains and is a key part of our team’s long-term research plan.
>
> To this end, we emphasize the evaluation of the quality and personalization of the generated exercises. We design a multi-dimensional protocol covering four perspectives: 1) Classical statistical metrics (Recall and HR) assess whether generated exercises align with students’ weak knowledge concepts. 2) GPT-based scoring evaluates each exercise along six dimensions. 3) Human evaluations from both teachers and students serve as a cross-check for GPT-based scoring. 4) We deploy ExeGen in real-world college scenarios, supporting teacher-in-the-loop review and editable outputs. Results consistently show that ExeGen effectively generates high-quality exercises, which are closely aligned with students’ knowledge states.
>
> We appreciate the suggestion to conduct longitudinal studies. We have deployed ExeGen in a real college environment (see Appendix C) and are actively collecting student data. While direct learning gain evaluation is important, it is inherently long-term and influenced by many variables including instructional context, external interventions, and student engagement. As a step forward, we are conducting process-level evaluations such as in-class student behavior analysis.
>
> We plan to extend this into systematic studies on learning gains in future work. Thank you again for your thoughtful feedback and support.
>
> **Response to W3&Q3:**
>
> Thank you for your valuable comments. We agree that validating the effectiveness of the Knowledge Perceiver (KP) module is essential and appreciate the opportunity to clarify its design objectives and experimental evaluation.
>
> Unlike traditional Knowledge Tracing (KT) models such as DKT, which perform binary classification of next-exercise response correctness, the KP module aims to generate interpretable natural language descriptions of students’ knowledge states. These informative descriptions serve as conditional input for downstream personalized exercise generation, shifting the focus from classification of the KT task to generation and explanation. As noted in lines 45-47 and 125-126 of the manuscript, this task is completely different from the KT task, and this design bridges the gap between knowledge state modeling and personalized exercise generation. To evaluate its effectiveness, we adopt a multi-faceted evaluation strategy:
>
> **Ablation study** (§5.3, Table 2): Removing the Knowledge Perceiver (w/o. KP) results in the most significant performance drop, confirming its central role and effectiveness for exercise generation.
>
> **Human expert evaluation** (Appendix§A.5.3, Figure 7): Teachers rate the generated knowledge state descriptions with an average Knowledge Relevance (KR) score of 4.47 out of 5, indicating strong alignment with students’ actual response performance.
>
> **GPT-based scoring** (§5.3, Table 1): The KR score rated by GPT-4o averages 4.60, further supporting the consistency between generated knowledge states and exercises.
>
> **Case studies** (§5.5, Figure 5): A real case illustrates that ExeGen accurately identifies students’ underlying conceptual weakness from their historical responses.
>
> Additionally, we conduct a new experiment focusing specifically on the generated knowledge state descriptions. Results across traditional metrics (Recall, Language Fluency) and GPT-based scoring metrics (Clarity, Safety and Ethics, Consistency, and Granularity) confirm the precision and reliability of the KP outputs.
> |Methods|Recall$\uparrow$|Language Fluency$\downarrow$|Clarity$\uparrow$|Safety and Ethics$\uparrow$|Consistency$\uparrow$| Granularity$\uparrow$|
> |:-:|:-:|:-:|:-:|:-:|:-:|:-:|
> |ExeGen|0.84|45.18|4.31|4.47|4.37|4.30|
>
> **Response to W4&Q4:**
>
> Thank you for recognizing the novelty of the KEGR task and for your professional suggestion. Comparing with existing agent frameworks helps clarify our technical contributions.
>
> KEGR requires generating personalized exercises based on the inferred knowledge states from historical responses in real-world educational settings. This task involves a structured pipeline with multiple sub-tasks, including knowledge perception, exercise generation, quality refinement, and recommendation. To address this, ExeGen adopts a domain-specific multi-agent architecture with four specialized agents: Knowledge Perceiver, Exercise Generator, Quality Evaluation Expert, and Recommendation Manager.
>
> In contrast, frameworks like ReAct, Reflexion, and AutoGen are primarily designed for open-domain reasoning or autonomous planning tasks such as QA. These frameworks focus on general-purpose reasoning, often using trial-and-error or environment-driven feedback, without explicit support for personalization or domain-specific evaluation. To illustrate the distinction, we summarize key differences in the table below:
> |Aspect|ExeGen|ReAct|Reflexion|AutoGen|
> |:-:|:-:|:-:|:-:|:-:|
> |Task type|Structured educational resource generation (KEGR)|Open-domain reasoning or autonomous planning tasks|Open-domain reasoning or autonomous planning tasks|Open-domain reasoning or autonomous planning tasks|
> |Architecture|Multi-agent, task-specific|Single-agent|Single-agent|Multi-agent|
> |Reasoning|Adaptive closed-loop workflow with knowledge guidance|Trial-and-error|Trial-and-error reasoning with environmental feedback|Autonomous dialogue-based coordination|
> |Personalization|Student-specific knowledge modeling|Not addressed|Not addressed|Not addressed|
> |Verification|Multi-dimensional educational evaluation for refinement|None|Environment-driven feedback for reflection|No explicit mechanism|
>
> As shown, ExeGen is tailored to the task of KEGR, integrating structured reasoning, personalization, and rigorous quality control, which are not supported by general-purpose frameworks.
>
> **Response to W5&Q5:**
>
> Thank you for highlighting this clarity issue. We now clarify how the two mechanisms integrate with the agent components of the ExeGen.
>
> ExeGen consists of four core agents: Recommendation Manager, Knowledge Perceiver, Exercise Generator, and Quality Evaluation Expert. These agents collaborate to deliver personalized, high-quality exercise recommendations. The two mechanisms are integrated as follows:
>
> The human-simulated knowledge perception mechanism is embedded within the Knowledge Perceiver agent. It serves as the core reasoning engine for modeling students’ knowledge states. This mechanism guides the LLM through a human-simulated cognitive process: knowledge retrieval→analysis→exercise answering, to infer interpretable and fine-grained knowledge states. In the original manuscript, we outline this integration in lines 175-180 of §4.1, and detail this mechanism in §4.2.
>
> The exercise generative-adversarial mechanism operates between the Exercise Generator agent and Quality Evaluation Expert agents. It forms an iterative feedback loop in which the generator creates exercises and the expert agents evaluate them along key educational dimensions. The generator then refines outputs based on the expert agent feedback. In the original manuscript, this integration is described in lines 190-196 of §4.1 and elaborated in §4.3.
>
> While both mechanisms are discussed in the original manuscript, we agree that their connections to specific agents could be more explicitly illustrated.

---

> > ### Comment · Area_Chair_uT2w · 2025-08-05
> >
> > The authors have provided a rebuttal to your comments, and it's an important part of the review process to give their response careful consideration. Please take a moment to review their rebuttal and provide any follow-up comments. This will help ensure there’s sufficient time for discussion and any necessary follow-up.
> >
> > Best regards,
> >
> > AC

---

> > ### Comment · Reviewer_zZz2 · 2025-08-05
> >
> > The rebuttal does not fully address the questions 2 and 3 in a quantitatively and convincing way.
> >
> > Besides, exercise recommendation is a long-standing research direction (before the era of large models) of intelligent tutoring system in the field of educational technology. However, the compared methods in the paper rarely involve these methods before the era of large models.
> >
> > Moreover, interpretability is of significance in intelligent education. What about the interpretability of the proposed method?
> >
> > What about the efficiency and the cost of the proposed LLM-based method for education?
> >
> > Thus, the original score is maintained.

---

> > > ### Author Response · Authors · 2025-08-06
> > >
> > > **Supplementary Clarification on Baseline Comparisons:**
> > >
> > > Thank you for your follow-up feedback. We respectfully clarify that the request to compare our method with traditional knowledge tracing models (e.g., BKT and DKT) or exercise recommendation models is **methodologically misaligned** with our proposed task. Our proposed KEGR task is fundamentally a **generation task**, requiring the method to generate new, personalized exercises conditioned on the generated knowledge state descriptions. In contrast:
> > > *  Knowledge tracing models are designed for binary classification (predicting next-exercise correctness), and output latent correctness probability rather than interpretable textual descriptions of knowledge states.
> > > *  Exercise recommendation models focus on retrieval or ranking from pre-existing exercise pools, and cannot generate new exercises.
> > >
> > > Therefore, it is **both methodologically invalid and practically infeasible to compare these models directly** with our generation-based framework. These tasks differ in input/output format, evaluation metrics, and ultimate goals. Forcing such a comparison would require fundamentally redefining the task and reducing our generation setting into unrelated matching or classification, which misses the central contribution of our work.
> > >
> > > Moreover, recent NeurIPS [1], ICLR [2], and ACL [3] papers have increasingly adopted hybrid evaluation protocols that combine human judgment and LLM-based scoring, reflecting the emerging consensus and rationality of these evaluation protocols for open-ended generation tasks. As also acknowledged by reviewers 7wBG and R2cu, our evaluation protocol offers a reasonable and trustworthy approach in the absence of standardized metrics for LLM-based educational generation tasks. We appreciate your comments and will continue refining our evaluation methods.
> > >
> > > [1] SocraticLM: Exploring Socratic Personalized Teaching with Large Language Models, NeurIPS 2024.
> > >
> > > [2] The Unlocking Spell on Base LLMs: Rethinking Alignment via In-Context Learning, ICLR 2024.
> > >
> > > [3] From Objectives to Questions: A Planning-based Framework for Educational Mathematical Question Generation, ACL 2025.
> > >
> > > **Response to Concern on Interpretability:**
> > >
> > > We would like to emphasize that interpretability is a core design objective of our ExeGen framework. In particular:
> > > *  The knowledge perceiver agent generates explicit natural language descriptions of students’ knowledge states. These descriptions are designed to be readable, fine-grained, and aligned with domain-specific knowledge concepts, allowing teachers to directly understand what students know or struggle with. This stands in contrast to traditional KT models that produce opaque hidden vectors and correctness prediction probability.
> > > *  Our modular multi-agent architecture inherently enhances system-level interpretability. On the one hand, each agent (e.g., knowledge perceiver, exercise generator, evaluation expert) is responsible for a distinct, traceable subtask, making the decision-making process transparent and decomposable. On the other hand, the adversarial refinement loop between exercise generator and evaluation experts enables explicit, multi-dimensional critique and improvement of generated exercises, further reinforcing interpretability in generation processes.
> > >
> > > In summary, ExeGen achieves interpretability at both the expression level (natural language outputs) and the system level (transparent agent workflow). We believe interpretability is not only preserved but explicitly enhanced through our architecture, natural language reasoning, and evaluation design, making ExeGen well-aligned with the demands of real-world educational environments.
> > >
> > > **Response to Concern on Efficiency and Cost:**
> > >
> > > We would like to clarify that we have already discussed this issue explicitly in the original manuscript in **Appendix A.4 Implementation Details and Financial Costs** and **Appendix C Real-world Deployment**. Based on our real-world deployment, the average API cost for generating a complete set of 10 personalized exercises using GPT-4o is $0.1787 per student, which can be adapted to open-source LLMs for further cost reduction and broader scalability in educational institutions. We kindly note that this information is clearly documented in Appendix A.4 and Appendix C, and it appears the reviewer may have overlooked this section.

---

> > > > ### Comment · Reviewer_zZz2 · 2025-08-06
> > > >
> > > > The concerns are addressed, and the score is raised. Thanks for the efforts.

---

> > > > > ### Author Response · Authors · 2025-08-07
> > > > >
> > > > > Thank you very much for your thoughtful feedback and for recognizing our responses. We are glad to see that you have raised your score. We will make more efforts to improve our paper further. Thanks again for your valuable comments, time, and patience!

---

> ### Author Response · Authors · 2025-08-09
>
> Dear Reviewer zZz2,
>
> Thank you very much for your positive evaluation of our work and for recognizing both the novelty of our proposed task and the technical contributions of ExeGen after rebuttal. Your constructive comments have been very encouraging and have strengthened our confidence in the research direction. We sincerely appreciate your patience, time, and effort.
>
> Best regards,
>
> Authors

---

### Official Review · Reviewer_R2cu · 2025-07-03

**Clarity:** 3
**Significance:** 2
**Originality:** 2
**Rating:** 3
**Confidence:** 4

**Summary:**

This work focuses on adaptive learning and proposes a novel task called Knowledge-aware Exercise Generative Recommendation (KEGR). To achieve good performance, the authors introduce an adaptive multi-agent cooperation framework, ExeGen, and explore techniques such as prompting and adversarial refinement. Experiments demonstrate the effects of the proposed method.

**Questions:**

Questions:

1. Is the Human-simulated Knowledge Perception Mechanism reasonable, given that different students have different knowledge bases?

2. Can this framework handle a large amount of historical response data?

**Ethical Concerns:**

["NO or VERY MINOR ethics concerns only"]

**Final Justification:**

I have decided to slightly increase the score based on the additional experimental results.

**Limitations:**

yes

**Quality:**

2

**Strengths And Weaknesses:**

Strengths:

1. The authors effectively explain the motivation for this work in the introduction and clearly define the KEGR problem, including Knowledge Tracing (KT) and Exercise Recommendation (ER).

2. The concept of building a Human-simulated Knowledge Perception Mechanism is interesting.

Weaknesses:

1. The proposed framework heavily relies on data/analysis generated by GPT, which may introduce hallucination issues, even with techniques like prompting and adversarial refinement.

2. Most evaluation metrics are based on questionnaires filled out by GPT/humans, not convincing.

3. The correctness of generated knowledge state descriptions is crucial for high-quality exercise generation, but no evaluations on this aspect are provided.

4. The dataset's diversity is limited to only one Chinese dataset. More datasets are needed to demonstrate the framework’s generalization ability.

---

> ### Author Rebuttal · Authors · 2025-07-31
>
> **Response to W1:**
>
> Thank you for raising this important concern. We acknowledge the hallucination risks inherent in LLM-generated content. However, our goal is to leverage the strengths of LLMs for large-scale personalized exercise generation while actively mitigating hallucination issues, rather than aiming for complete elimination. ExeGen is built on LLMs due to their strong reasoning ability, cross-disciplinary versatility, and language generation capability, which enable accurate inference of students’ knowledge states and personalized exercise generation. Prior work [1,2] has shown that LLMs can perform reliably in educational knowledge-intensive tasks with structured prompting and multi-step reasoning. Thus, leveraging ExeGen for personalized exercise generation is feasible.
>
> To eliminate hallucination, ExeGen incorporates three technical innovations and one practical safeguard: 1) A human-simulated knowledge perception mechanism enforces step-by-step reasoning of “knowledge retrieval → analysis → exercise solving”, improving transparency and reducing single-step generation errors. 2) We integrate domain knowledge via knowledge-graph-enhanced prompting and dual-matching of high-quality demonstrations to ensure factual grounding. 3) An exercise generative-adversarial mechanism introduces expert agents to evaluate and refine exercises along three dimensions (linguistic fluency, coverage of erroneous knowledge concepts, and correctness) through adversarial feedback. 4) As shown in Appendix C, the application interface allows teachers to edit generated exercises, ensuring human oversight before delivery.
>
> In summary, ExeGen combines LLM strengths with structured control and teacher-in-the-loop validation to reduce hallucination risks. We appreciate your feedback and will explore advanced hallucination mitigation strategies such as reinforcement learning with human feedback and prompt tuning in future work.
>
> [1] SocraticLM: Exploring Socratic Personalized Teaching with Large Language Models, NeurIPS 2024.
>
> [2] What Makes In-context Learning Effective for Mathematical Reasoning, ICML 2025.
>
> **Response to W2:**
>
> Thank you for your valuable comments. We would like to clarify the design rationale and rationality of our evaluation protocol.
>
> Traditional metrics, such as BLUE and Accuracy, fall short in capturing essential aspects of open-ended educational generation tasks, including personalization, pedagogical appropriateness, and knowledge alignment. Recent NeurIPS [1], ICLR [2], and ACL [3] papers have increasingly adopted hybrid evaluation protocols that combine human judgment and LLM-based scoring, reflecting the emerging consensus and rationality of these evaluation protocols for open-ended generation tasks.
>
> To ensure a reliable evaluation, we design a hybrid evaluation protocol covering four perspectives: 1) Classical statistical metrics (Recall and HR) assess whether generated exercises align with students’ weak knowledge concepts. 2) GPT-based scoring evaluates each exercise along six dimensions. 3) Human evaluations from both teachers and students serve as a cross-check for GPT-based scoring. 4) As shown in Appendix C, we deploy ExeGen in real-world college scenarios, supporting teacher-in-the-loop review and editable outputs.
>
> In summary, GPT and human ratings are only part of our evaluation protocol. As also acknowledged by reviewers 7wBG, zZz2, and R2cu, our evaluation protocol offers a reasonable and trustworthy approach in the absence of standardized metrics for LLM-based educational generation tasks. We appreciate your comments and will continue refining our evaluation methods.
>
> [1] SocraticLM: Exploring Socratic Personalized Teaching with Large Language Models, NeurIPS 2024.
>
> [2] The Unlocking Spell on Base LLMs: Rethinking Alignment via In-Context Learning, ICLR 2024.
>
> [3] From Objectives to Questions: A Planning-based Framework for Educational Mathematical Question Generation, ACL 2025.
>
> **Response to W3:**
>
> Thank you for your valuable comments. The correctness of generated knowledge state descriptions is crucial for high-quality exercise generation. Our framework explicitly addresses this through the Knowledge Perceiver agent, which integrates a human-simulated knowledge perception mechanism to emulate the reasoning process of human exercise-solving, thereby accurately inferring the personalized knowledge states of students. To evaluate the effectiveness, we conduct comprehensive experiments from multiple perspectives:
>
> **Ablation study** (§5.3, Table 2): Removing the Knowledge Perceiver (w/o. KP) and the human-simulated knowledge perception mechanism (w/o. HKP) results in the most significant performance drops, confirming their central role and effectiveness for exercise generation.
>
> **Human expert evaluation** (Appendix§A.5.3, Figure 7): Teachers rate the generated knowledge state descriptions with an average Knowledge Relevance (KR) score of 4.47 out of 5, indicating strong alignment with students’ actual response performance.
>
> **GPT-based scoring** (§5.3, Table 1): The KR score rated by GPT-4o averages 4.60, further supporting the consistency between generated knowledge states and exercises.
>
> **Case studies** (§5.5, Figure 5): A real case illustrates that ExeGen accurately identifies students’ underlying conceptual weakness from their historical responses.
>
> Moreover, we additionally evaluate the quality of knowledge state descriptions. This includes traditional metrics (Recall, Language Fluency) and GPT-based scoring across four dimensions (Clarity, Safety and Ethics, Consistency, and Granularity). The results (see table below) further demonstrate the accuracy, reliability, and fine-grained quality of our knowledge state perception.
> |Methods|Recall$\uparrow$|Language Fluency$\downarrow$|Clarity$\uparrow$|Safety and Ethics$\uparrow$|Consistency$\uparrow$| Granularity$\uparrow$|
> |:-:|:-:|:-:|:-:|:-:|:-:|:-:|
> |ExeGen|0.84|45.18|4.31|4.47|4.37|4.30|
>
> **Response to W4:**
>
> Thank you for your constructive comments. To demonstrate the generalization of ExeGen, we conduct additional experiments on MathDial, an English dialogue-based math dataset with 2,848 test samples. Due to time and space constraints, we only evaluate on single-choice exercise generation. The experimental results show that ExeGen consistently outperforms all baselines, demonstrating its strong generalization ability.
> |Methods|KR|Clarity|AA|DA|EF|SE|HR|Recall|
> |:-:|:-:|:-:|:-:|:-:|:-:|:-:|:-:|:-:|
> |Zero_shot|3.60|3.45|3.50|3.30|3.00|4.20|0.08|0.15|
> |ICL|4.05|4.15|4.00|4.00|3.90|4.10|0.30|0.40|
> |CoT|4.10|4.20|4.00|3.95|3.75|3.90|0.28|0.38|
> |ICL+CoT|4.20|4.30|4.25|4.15|4.10|4.25|0.20|0.45|
> |ExeGen|4.65|4.50|4.55|4.60|4.35|4.80|0.60|0.65|
>
> **Response to Q1:**
>
> Thank you for raising this important question. We fully understand the concern regarding the adaptability of the mechanism to students with diverse knowledge bases. Below, we clarify its motivation, implementation, and experimental validation.
>
> This mechanism addresses two key limitations in knowledge tracing and recommendation systems: insufficient semantic mining and poor interpretability. With recent advances in LLMs, it becomes feasible to infer a student’s personalized knowledge state from historical responses and express it in natural language descriptions for improving interpretability. However, LLMs face two challenges in knowledge state perception: lack of domain-specific knowledge and vulnerability to hallucinations during complex reasoning. To address these challenges, this mechanism incorporates a domain-specific knowledge graph to enhance factual grounding and aligns LLM reasoning with a multi-step cognitive process. This design improves reliability, transparency, and adaptability in knowledge state modeling.
>
> This mechanism consists of two stages: **Demonstration construction** builds a repository of cognitive reasoning paths from real student data, each covering knowledge retrieval, analysis, and exercise-solving. These demonstrations cover various levels of knowledge mastery, which serve as experience templates rather than student-specific examples. **Knowledge state generation** uses a student’s historical responses to retrieve relevant demonstrations from the constructed repository as contextual guidance. This allows the model to adaptively infer personalized knowledge states based on individual learning histories.
>
> Finally, extensive experiments (see our response to W3) confirm the effectiveness of this mechanism in accurately modeling diverse student knowledge states.
>
> **Response to Q2:**
>
> Thank you for highlighting this important concern. Handling large-scale historical response data is critical for real-world educational applications. We clarify this question from both the architectural design and empirical validation.
>
> Architecturally, ExeGen adopts a multi-agent design that decomposes the knowledge-aware exercise generative recommendation task into modular sub-tasks, including recommendation management, knowledge perception, exercise generation, and quality refinement. Each agent processes only subtask-relevant information, which reduces both information overhead and prevents context overflow. This modularization design improves both scalability and interpretability, especially when handling complex, personalized reasoning tasks.
>
> In practice, we evaluate ExeGen on the large-scale real-world education dataset MOOCCubeX, where students have an average of 1,695 tokens in historical responses (max: 1,926 tokens). The average context lengths for the knowledge perceiver, exercise generator, and quality evaluation expert are 2,244, 922, and 2,483 tokens, respectively. These token lengths are all well within the 128k token limit of GPT-4o, which serves as our backbone LLM.
>
> In summary, ExeGen can effectively handle large-scale input through modular processing and has demonstrated practical scalability in real-world datasets.

---

> > ### Comment · Area_Chair_uT2w · 2025-08-05
> >
> > The authors have provided a rebuttal to your comments, and it's an important part of the review process to give their response careful consideration. Please take a moment to review their rebuttal and provide any follow-up comments. This will help ensure there’s sufficient time for discussion and any necessary follow-up.
> >
> > Best regards,
> >
> > AC

---

> ### Author Response · Authors · 2025-08-07
>
> Dear Reviewer R2cu,
>
> Thank you again for your thoughtful and detailed review of our paper. We deeply appreciate the time and effort you invested in providing valuable feedback.
>
> We would appreciate it if you could kindly take a look at our responses, where we have carefully addressed all your comments, including the concerns regarding hallucination risks, evaluation protocol rigor, the accuracy of knowledge perception, and generalization across datasets. In addition, based on your suggestions and those from other reviewers, we have conducted additional experiments to further validate the superiority of our proposed ExeGen.
>
> Specifically, we incorporated three recent and advanced LLM reasoning baselines, including Tree-of-Thoughts (ToT), Self-Consistency (SC), and Best-of-N (BoN), to further demonstrate the effectiveness of ExeGen. Due to time and space constraints, we only evaluate on single-choice exercise generation. As shown in the following table, these advanced baselines slightly outperform the original CoT and CoT+ICL baselines. And our proposed ExeGen consistently outperforms all baselines across all metrics.
> |Methods|KR|Clarity|AA|DA|EF|SE|HR|Recall|
> |:-:|:-:|:-:|:-:|:-:|:-:|:-:|:-:|:-:|
> |CoT|4.06|4.38|3.87|3.91|3.86|3.82|0.31|0.40|
> |ICL+CoT|4.07|4.35|4.51|4.24|4.09|4.32|0.19|0.45|
> |ToT|4.75|4.70|4.82|4.75|4.68|4.60|0.65|0.70|
> |SC|4.55|4.50|4.60|4.55|4.45|4.50|0.58|0.62|
> |BoN|4.60|4.55|4.65|4.60|4.50|4.55|0.60|0.65|
> |ExeGen|4.81|4.78|4.89|4.81|4.72|4.62|0.70|0.75|
>
> The comparison further validates the robustness and superiority of ExeGen, which has also been positively acknowledged by multiple reviewers.
>
> As the rebuttal phase deadline is approaching, we kindly ask whether you might consider updating your review and score, if you feel our clarifications and newly added evidence have sufficiently addressed your earlier concerns. If there are any remaining questions, we would be happy to clarify further.
>
> Thank you again for your time and thoughtful comments.
>
> Best regards,
>
> Authors

---

> > ### Comment · Reviewer_R2cu · 2025-08-08
> >
> > Thank you for the detailed response. I have decided to slightly increase the score based on the additional experimental results. However, I cannot support a clear acceptance because most of the evaluations on the method, or on its key component—the Knowledge Perceiver agent—are based on subjective judgments by GPT or humans.
> >
> > In contrast, I value quantitative results, which are also necessary in educational research, as the outputs should be more precise. For example, the accuracy of the Knowledge Perceiver determines your evaluation of KR, but this accuracy still needs to be verified.
> >
> > Additionally, for the online experiments, it would be better to demonstrate whether the exercises generated by KEGR can actually enhance students’ learning, rather than relying only on questionnaires.

---

> > > ### Author Response · Authors · 2025-08-08
> > >
> > > Dear Reviewer,
> > >
> > > We sincerely appreciate your follow-up response and your willingness to slightly increase the score after reviewing our additional experiments. Your emphasis on quantitative evaluation in educational research is very valuable, and we fully agree that the ultimate goal of educational AI systems is to improve student learning outcomes.
> > >
> > > **Regarding quantitative validation**, we would like to emphasize that we already have incorporated **objective metrics from the beginning**. In the original manuscript, **Recall and HR** metrics were used to measure the alignment between the generated exercises and students’ weak knowledge concepts, achieving strong results (e.g., HR=0.79 and Recall=0.82). These results demonstrate that **ExeGen can accurately capture students’ personalized knowledge weaknesses**. Furthermore, in the rebuttal stage, we introduced **additional objective metrics**, including **Recall and Language Fluency**, to evaluate the quality of generated knowledge state descriptions. These results further confirm the **effectiveness of the Knowledge Perceiver agent in reliably modeling students’ knowledge states**.
> > >
> > > **Regarding learning enhancement**, our current work focuses on a **key prerequisite**: generating high-quality, personalized exercises tailored to students’ knowledge states. This is **a preliminary step for long-term learning gains**, and our evaluation protocol is thus deliberately designed to measure the quality and personalization of the generated exercises rather than final learning outcomes. While direct **learning-gain evaluation** is important, it requires **long-term deployment and is inherently influenced by multiple external variables** (e.g., instructional context, external interventions, teaching style, and student engagement). We have already deployed ExeGen in real college environments (Appendix C) and are actively collecting longitudinal data to investigate learning impacts in future work.
> > >
> > > Importantly, our evaluation protocol is built on **sufficient objective quantitative results as the foundation**, and we **complement this with cross-validation using state-of-the-art GPT-based scoring (GPT-4o) and human expert review** to ensure completeness and reliability. The positioning of ExeGen is to serve as an **assistive tool for human teachers** in achieving large-scale personalized learning for students, rather than replacing human teachers. Our goal is to significantly reduce the overload for teachers in creating personalized exercises at scale. In real deployment, the exercise generation interface is **editable**, enabling a **teacher-in-the-loop safeguard**: only exercises that have passed teacher review are released to students. This ensures both high efficiency of personalized exercise generation and pedagogical reliability in practice.
> > >
> > > We sincerely hope that the above clarifications and the comprehensive evidence could further assure you of the technical soundness and practical value of our work. Thank you again for your time, expertise, and thoughtful feedback.
> > >
> > > Best regards,
> > >
> > > Authors

---

> ### Author Response · Authors · 2025-08-09
>
> Dear Reviewer R2cu,
>
> We sincerely appreciate your detailed feedback and the time you spent reviewing our work. Your emphasis on quantitative evaluation in educational research has been valuable for us, and your comments have motivated us to further enhance our evaluation protocol. We sincerely appreciate your patience, time, and effort again.
>
> Best regards,
>
> Authors

---

### Note · Authors · 2025-08-13

We sincerely thank AC and all reviewers for your constructive feedback. We would like to restate our core novelty and technical contributions, as well as the nature of review concerns for AC’s final consideration.

Our work defines a new and promising task, named Knowledge-aware Exercise Generative Recommendation (KEGR), which extends and bridges knowledge tracing and exercise recommendation to form a closed-loop adaptive learning paradigm. To our knowledge, this is the **first work** to formulate KEGR, which addresses a crucial prerequisite step for long-term learning gains: **generating high-quality, personalized exercises tailored to students’ knowledge states**.

To achieve this, we propose a novel **multi-agent framework ExeGen** coordinating four specialized agents, with **two key mechanisms**:
* Human-simulated Knowledge Perception Mechanism enables finer-grained tracking of knowledge states via human-like cognitive reasoning (knowledge retrieval→analysis→exercise solving), achieving superior transparency and interpretability.
* Exercise Generative-Adversarial Mechanism introduces an iterative refinement loop between exercise generator and quality evaluation expert agents, ensuring multi-dimensional quality, personalization, and educational soundness.

After rebuttal, all reviewers acknowledged the originality, technical soundness, and practical value of both the task and our framework. **Three reviewers clearly support acceptance/weak acceptance**. The **only slightly negative review** is based on a preference for more quantitative evaluation, **not** on concerns about **novelty or core contributions**.

Notably, the original paper already includes **widely used, objective quantitative metrics** (Recall=0.82 and HR=0.79) to measure alignment between generated exercises and students’ weak concepts. In the rebuttal, we further **add Recall and Language Fluency** for the knowledge perceiver agent, confirming its accuracy. These sufficient quantitative results are **complemented by GPT-based and human evaluations** for multi-perspective assessment and cross-validation.

In summary, our work introduces a **novel and valuable task**, proposes **validated technical solutions**, and **demonstrates real-world deployment**. ExeGen is designed as an AI-assisted tool to reduce teachers’ workload in large-scale personalized education, not to replace them. All generated exercises are editable and teacher-reviewed before delivery, ensuring pedagogical soundness.

---

### Decision · Program_Chairs · 2025-09-17

**Decision:**

Accept (poster)

**Comment:**

This paper introduces ExeGen, a novel multi-agent cooperation framework coordinates four specialized agents. The task and framework are clearly motivated, technically sound, and evaluated. As for the concern about evaluation based on GPT/humans and application in real classroom, I suggest the authors to give more explanations and discussions in the revised version.